# Selective retention of dysfunctional mitochondria during asymmetric cell division in yeast

Xenia Chelius[1], Veronika Bartosch[1], Nathalie Rausch[1], Magdalena Haubner[1], Jana Schramm[1], Ralf J. Braun[2], Till Klecker[1]*, Benedikt Westermann[1]*

1 Zellbiologie, Universität Bayreuth, Bayreuth, Germany, 2 Department Medizin, Fakultät Medizin/Zahnmedizin, Danube Private University, Krems, Austria

* till.klecker@uni-bayreuth.de (TK); benedikt.westermann@uni-bayreuth.de (BW)

## Abstract

Decline of mitochondrial function is a hallmark of cellular aging. To counteract this process, some cells inherit mitochondria asymmetrically to rejuvenate daughter cells. The molecular mechanisms that control this process are poorly understood. Here, we made use of matrix-targeted D-amino acid oxidase (Su9-DAO) to selectively trigger oxidative damage in yeast mitochondria. We observed that dysfunctional mitochondria become fusion-incompetent and immotile. Lack of bud-directed movements is caused by defective recruitment of the myosin motor, Myo2. Intriguingly, intact mitochondria that are present in the same cell continue to move into the bud, establishing that quality control occurs directly at the level of the organelle in the mother. The selection of healthy organelles for inheritance no longer works in the absence of the mitochondrial Myo2 adapter protein Mmr1. Together, our data suggest a mechanism in which the combination of blocked fusion and loss of motor protein ensures that damaged mitochondria are retained in the mother cell to ensure rejuvenation of the bud.

**Data Availability Statement:** All relevant data are within the paper and its Supporting Information files.

## Introduction

Mitochondria cannot be made de novo but have to be passed on to daughter cells during cell division. Depending on the organism and cell type, mitochondrial inheritance involves stochastic or ordered partitioning strategies [1–4]. In many mammalian cell types, mitochondria fragment during mitosis. This activity in concert with cytoskeleton-dependent motility contributes to stochastic partitioning of mitochondria to the daughter cells [5–7]. However, certain cell types divide asymmetrically and actively control uneven distribution of their cell organelles. This results in distinct fates of progeny cells and is important, e.g., for the maintenance of stem cells in metazoans or as a defense against aging in microorganisms [8,9].

Decline of mitochondrial function and accumulation of mutations in the mitochondrial DNA (mtDNA) are hallmarks of aging [10,11]. Therefore, the proper distribution of intact mitochondria during cell division is crucial for cell homeostasis, and their asymmetric inheritance is thought to contribute to the rejuvenation of progeny cells. An analysis of stem-like cells revealed that young mitochondria are preferentially apportioned to daughter cells that

**Funding:** This work was funded by the Deutsche Forschungsgemeinschaft, project numbers 433461293 (to BW) and 491183248 (Open Access Publishing Fund of the University of Bayreuth), and by Elitenetzwerk Bayern through the "Biological Physics" program (to BW). The funders had no role in study design, data collection and analysis, decision to publish, or preparation of the manuscript.

**Competing interests:** The authors have declared that no competing interests exist.

**Abbreviations:** 5-FOA, 5-fluoroorotic acid; DAO, D-amino acid oxidase; DIC, differential interference contrast; mtDNA, mitochondrial DNA; mtERFP, mitochondria-targeted enhanced red fluorescent protein; mtGFP, mitochondria-targeted GFP; PI, propidium iodide; RACF, retrograde actin cable flow; ROI, region of interest; ROS, reactive oxygen species; SCD, synthetic complete dextrose medium; SD, standard deviation; Su9-DAO, mitochondrial matrix-targeted D-amino acid oxidase; TMRM, tetramethylrhodamine methyl ester; yEGFP, yeast enhanced GFP.

maintain stem cell traits, whereas daughter cells destined to differentiate receive a larger share of old mitochondria [12]. Thus, it appears that young and healthy mitochondria are preferentially apportioned to the daughter cell that has to maintain a high replicative potential.

Budding yeast *Saccharomyces cerevisiae* is a powerful model organism to study the asymmetric inheritance of organelles [13–16]. Each cell division produces a virgin daughter cell that buds off from its aging mother. During replicative aging, the mother accumulates aging factors and dies after about 20 to 30 cell divisions. Remarkably, daughter cells are born with full replicative potential, even if they bud off from old mother cells. To ensure rejuvenation of their daughters, yeast mother cells retain protein aggregates, damaged organelles, and other harmful materials and avoid their transport into the bud [17–19].

Several observations suggest that mitochondria are asymmetrically inherited in yeast [13,20]. A mutation in the *ATP2* gene, encoding a subunit of the $F_1$ part of the mitochondrial ATP synthase, leads to loss of mother-daughter age asymmetry and accumulation of dysfunctional mitochondria [21]. Aconitase, an enzyme of the citric acid cycle, is asymmetrically distributed in mitochondria of aging mothers and their daughters [22]. Fluorescent biosensors revealed that mitochondria in the bud have lower levels of reactive oxygen species (ROS) and are more reducing than their counterparts in the mother [23,24]. The mitochondria-to-cell size ratio is tightly controlled during inheritance in the bud, whereas it continually decreases in aging mothers [25]. Furthermore, protein aggregates in the mitochondrial matrix are sequestered to a specific deposit site, which is retained in mother cells [26].

During the cell cycle, mitochondria enter the bud immediately upon its emergence and exhibit continuous anterograde and retrograde movements concomitant with frequent fusion and fission [27,28]. Mitochondria are transported along actin cables towards the bud by the myosin motor, Myo2 [29–31]. Binding of Myo2 to mitochondria is promoted by Ypt11, a small rab GTPase, and Mmr1, a protein peripherally associated with the mitochondrial outer membrane [29,32–34]. Δ*mmr1* and Δ*ypt11* single mutants have only slight mitochondrial inheritance defects while the Δ*mmr1* Δ*ypt11* double mutant is inviable [33,34] or severely sick [35]. Simultaneous loss of active Mmr1 and Ypt11 results in a synthetic mitochondrial inheritance defect [33–36] indicating that Ypt11 and Mmr1 have partially redundant functions in Myo2 recruitment. Mitochondrial cortex tethers ensure that a part of the mitochondrial network is retained in the mother cell upon cytokinesis [37–40]. Thus, multiple activities of the transport, retention, fusion, and fission machineries have to be orchestrated to promote ordered mitochondrial inheritance [41–43].

The mechanisms governing asymmetric distribution of mitochondria between the mother cell and the bud are poorly understood. Here, we have investigated how oxidative damage affects the inheritance of mitochondria during asymmetric division of yeast cells. We find that ROS produced in mitochondria trigger selective fragmentation and retention of dysfunctional organelles in the mother, while healthy mitochondria that are present in the same cell continue to move towards the bud. We provide evidence that the selection of fit mitochondria for inheritance occurs directly at the organellar level. Our results demonstrate that selection of mitochondria for bud-directed transport is achieved by Myo2 motor recruitment and Mmr1 plays a crucial role in this process.

## Results

### Mitochondria reversibly fragment and become immobile during $H_2O_2$ stress

ROS, such as superoxide and hydrogen peroxide ($H_2O_2$), are inevitable byproducts of oxidative phosphorylation in mitochondria [44]. While ROS have important physiological functions

and act as second messengers in signaling pathways, it is also clear that excess levels generated by damaged mitochondria may become toxic to the cell [45–47].

To test how yeast mitochondria react to excess $H_2O_2$, we incubated wild-type cells expressing mitochondria-targeted GFP (mtGFP) for 15 min in medium containing 0.3 or 0.5 mM $H_2O_2$ and analyzed them by fluorescence microscopy. Consistent with earlier reports [48–51], we observed excessive fragmentation of mitochondria (Fig 1A and 1B). Propidium iodide (PI) staining confirmed that cells were still alive under these conditions (Figs 1C and S1A). We observed similar mitochondrial fragmentation upon treatment with diamide, a thiol oxidizing agent (S1B and S1C Fig). This suggests that mitochondrial fragmentation is a general response to oxidative stress.

To analyze mitochondrial motility under ROS stress, we examined mitochondria of $H_2O_2$-treated yeast cells by time-resolved fluorescence microscopy. Mitochondrial fragmentation occurred within the first 15 min of oxidative stress, and mitochondria became almost completely immobile (S1 and S2 Videos). To test whether these effects are reversible, we allowed stressed cells to recover in fresh medium and then analyzed mitochondrial morphology again. Recovered cells were indiscernible from non-stressed control cells, both after $H_2O_2$ and diamide treatment (Figs 1A, 1B, S1B and S1C and S3 Video). Thus, oxidative stress induces rapid fragmentation and immobilization of mitochondria, which is fully reversible when the stress is relieved.

It has been reported that treatment of yeast cells with $H_2O_2$ or diamide induces a transient depolarization of the actin cytoskeleton [52]. Therefore, we subjected cells to $H_2O_2$ stress and analyzed them by staining with rhodamine phalloidin. Non-stressed cells showed the typical appearance of actin cables extending from the bud neck into the mother cell and actin patches located mainly in the bud (Fig 1D and 1E). Consistent with earlier observations [52], we found that actin cables were largely lost upon oxidative stress and actin patches were evenly distributed between the mother cell and bud (Fig 1D and 1E).

Actin cables serve as tracks for transport of mitochondria and other organelles into the bud. Organelle transport is powered by the myosin motor Myo2 [15,42]. In mitotically growing cells, Myo2 is located mainly at the bud tip or in large budded cells at the bud neck [53]. We found that Myo2-GFP was diffusely distributed in the cytoplasm of most stressed cells (Fig 1F and 1G). This suggests that exogenously added $H_2O_2$ triggers a general stress response that affects the actin cytoskeleton and all Myo2-dependent transport processes. This will not only impair the motility of mitochondria, but presumably also that of peroxisomes, vacuoles, secretory vesicles, and other Myo2 cargoes.

## Mitochondria-targeted D-amino acid oxidase affects mitochondrial morphology and fusion

Next, we employed a genetically encoded ROS stressor to analyze mitochondrial behavior in response to elevated $H_2O_2$ levels in the mitochondrial matrix. D-amino acid oxidase (DAO) is an enzyme that deaminates D-amino acids and concomitantly reduces FAD. Subsequent re-oxidation of $FADH_2$ by molecular oxygen produces $H_2O_2$ [54]. Heterologous expression of DAO can be used to induce D-alanine-controlled production of $H_2O_2$ in cells [55,56]. To induce $H_2O_2$ stress in the mitochondrial matrix, we used the previously described Su9-DAO construct [57]. It consists of the mitochondrial presequence of subunit 9 of the ATP synthase of *Neurospora crassa* fused to the DAO enzyme from *Rhodotorula gracilis* expressed from the constitutive *TEF* promoter. Su9-DAO was shown to produce deleterious levels of $H_2O_2$ in mitochondria of yeast and mammalian cells upon addition of D-alanine, but not L-alanine [57,58].

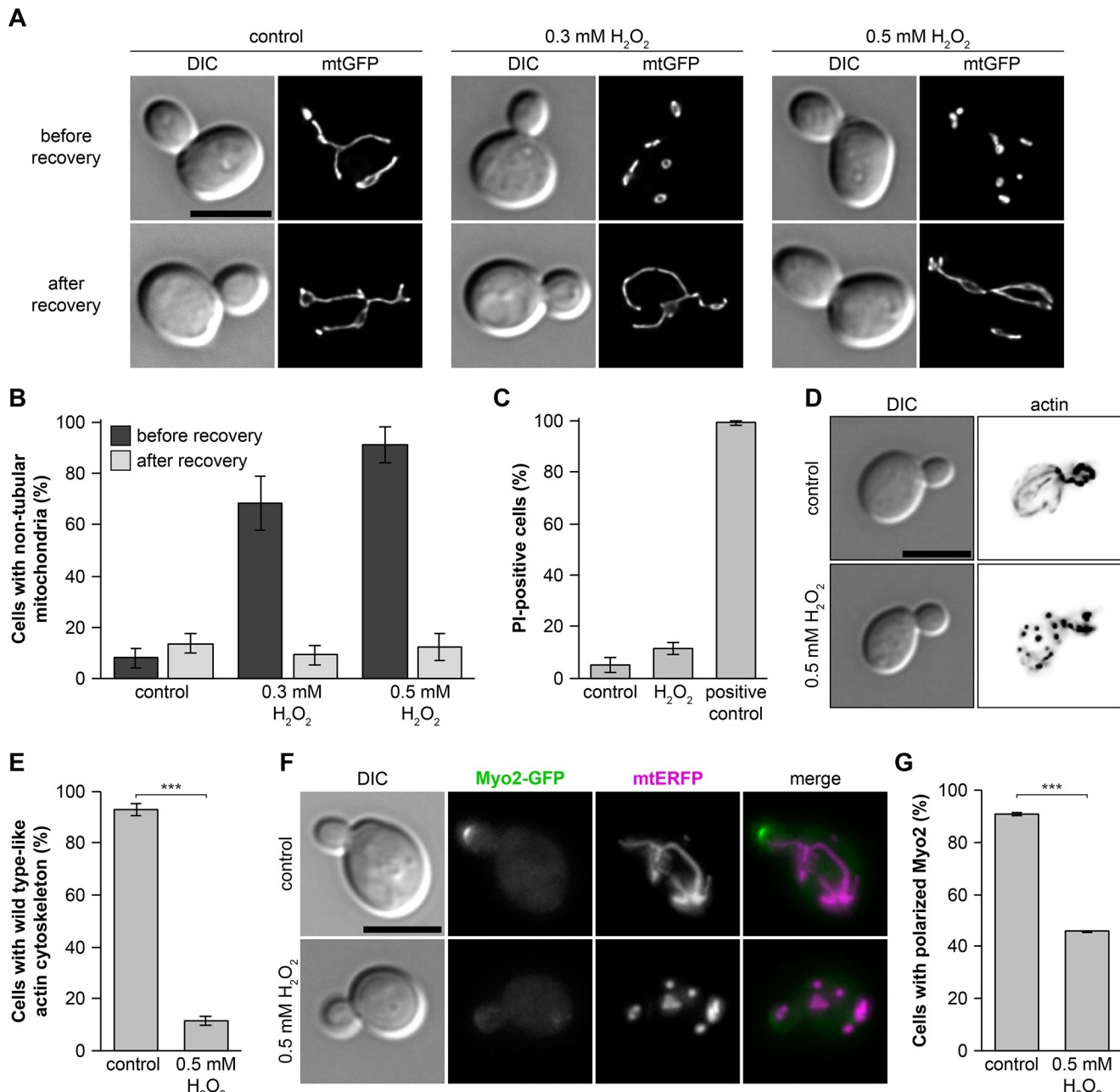

**Fig 1. H₂O₂ treatment induces reversible fragmentation of mitochondria and disappearance of actin cables. (A and B)** Yeast cells expressing mtGFP were treated with exogenously added $H_2O_2$ for 15 min and analyzed by DIC and fluorescence microscopy. For recovery, cells were incubated for 90 min in fresh medium. Fluorescence images are z stacks subjected to deconvolution. Mitochondrial morphology was quantified in 150 cells per sample (triplicate experiments ± SD). **(C)** Cells were treated with 0.5 mM $H_2O_2$ as in (A) and stained with PI. Cells that were killed at 100°C served as a positive control. PI-positive cells were quantified by fluorescence microscopy analysis in 200 cells per sample (triplicate experiments ± SD); see S1A Fig for representative images. **(D and E)** Cells were treated as in (A) and stained with rhodamine phalloidine. Fluorescence images are z stacks subjected to deconvolution. Actin morphology was quantified in at least 100 cells per sample (triplicate experiments ± SD). **(F and G)** Yeast cells expressing Myo2-GFP and mtERFP were treated with exogenously added $H_2O_2$ for 15 min. Fluorescence images are maximum intensity projections of z stacks. Myo2-GFP localization was quantified in at least 100 cells per sample (triplicate experiments ± SD). Bars, 5 μm. Data pooling and statistics are detailed in S2 Table. The data underlying this figure can be found in S1 Datasheet. DIC, differential interference contrast; mtERFP, mitochondria-targeted enhanced red fluorescent protein; mtGFP, mitochondria-targeted GFP; PI, propidium iodide; SD, standard deviation.

We integrated the *Su9-DAO* coding sequence into the genome and observed that growth of Su9-DAO-expressing cells on agar plates was impaired by the addition of D-alanine in a dose-dependent manner. Growth was not affected by L-alanine (Fig 2A). This suggests that Su9-DAO produces $H_2O_2$ in sufficient amounts to become toxic to the cells. We also observed that cells were more sensitive to Su9-DAO activity when grown on non-fermentable carbon sources (Fig 2A), suggesting that mitochondrial respiration is particularly affected.

To confirm that $H_2O_2$ production by Su9-DAO compromises mitochondrial function, we stained yeast cells with tetramethylrhodamine methyl ester (TMRM), a membrane potential-sensitive dye. Incubation of Su9-DAO-expressing cells for 3 h in D-alanine-containing medium was sufficient to break down the mitochondrial membrane potential. These cells were indiscernible from respiratory-deficient *rho⁰* cells lacking mtDNA (Figs 2B and S2A). We conclude that DAO activity in the mitochondrial matrix severely impairs mitochondrial function.

Fluorescence microscopy of Su9-DAO-expressing cells grown in liquid culture revealed that mitochondria became highly fragmented and aggregated when mitochondrial $H_2O_2$ production was induced by the addition of D-alanine to the medium (Fig 2C and 2D and S4 and S5 Videos). PI staining confirmed that cells were viable under these conditions (S2B Fig), and mitochondrial morphology was not affected by D-alanine in strains lacking the Su9-DAO construct (S2C and S2D Fig). Thus, production of $H_2O_2$ in the mitochondrial matrix induces mitochondrial fragmentation under nonlethal conditions, similar to exogenously added $H_2O_2$.

The respiratory chain is thought to constitute the major source of mitochondrial ROS [44]. To test whether it is also required for $H_2O_2$-induced mitochondrial fragmentation, we examined growth and mitochondrial morphology in Su9-DAO expressing Δ*mip1* cells lacking the mitochondrial DNA polymerase. This mutant is respiratory deficient as it lacks the mtDNA-encoded respiratory chain subunits. Growth impairment and mitochondrial fragmentation in Δ*mip1* cells were as severe as in wild-type cells (Fig 2E–2G), indicating that respiratory chain activity is not required.

Fragmentation of mitochondria under ROS stress suggests that mitochondrial fusion might be compromised. To test this, we assayed mitochondrial fusion in zygotes. Haploid cells of opposite mating types containing mitochondria labeled with either GFP or ERFP were grown in L-alanine or D-alanine-containing media, mixed to allow mating and zygote formation, and analyzed by fluorescence microscopy. Zygotes obtained from cells grown in L-alanine-containing media showed almost complete mixing of mitochondrial labels, indicating efficient mitochondrial fusion. In contrast, upon incubation in D-alanine-containing media only about 20% of zygotes showed fused mitochondria when both mating partners contained Su9-DAO, and 40% when a Su9-DAO expressing strain was mated with a wild type lacking this construct (Figs 2H, 2I, and S2E). We conclude that mitochondrial ROS stress severely impairs mitochondrial fusion.

## Mitochondrially produced $H_2O_2$ impairs mitochondrial motility and inheritance

The Su9-DAO construct allowed us to induce ROS stress selectively in mitochondria under sublethal conditions. We took advantage of this system to test whether $H_2O_2$ production in mitochondria impairs motility and inheritance of the organelle. Time-resolved fluorescence microscopy revealed that directed mitochondrial movements in Su9-DAO-expressing cells ceased about 2 to 3 h after addition of D-alanine (S4 and S5 Videos). Furthermore, we quantified the presence of mitochondria in medium-sized buds. About 80% of the control cells incubated with L-alanine showed efficient mitochondrial inheritance. In contrast, only about 20% of the buds contained a normal amount of mitochondria in the presence of D-alanine.

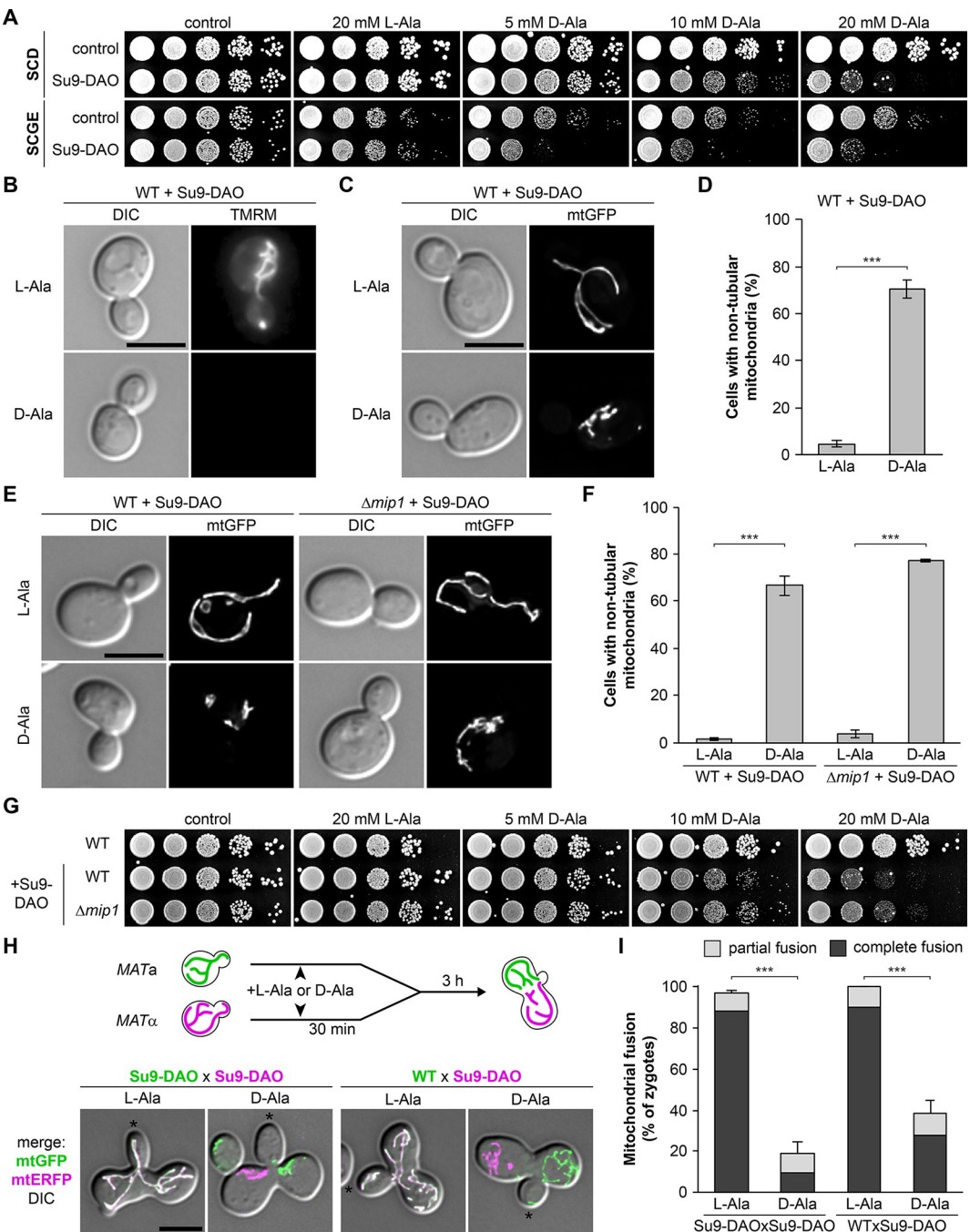

**Fig 2. Mitochondrially produced H$_2$O$_2$ induces loss of membrane potential, mitochondrial fragmentation, and block of fusion. (A)** Ten-fold serial dilutions of wild-type cells without (control) and with Su9-DAO were spotted on agar plates containing minimal complete medium with glucose (SCD) or glycerol and ethanol (SCGE) as carbon source and incubated at 30˚C. **(B)** Wild-type cells expressing Su9-DAO were incubated in the presence of L-alanine or D-alanine for 3 h and then stained with TMRM. Fluorescence microscopy images are maximum intensity projections of z stacks using identical camera settings. Additional images are shown in S2A Fig. **(C and D)** Wild-type cells expressing Su9-DAO and mtGFP were incubated for 3 h in the presence of L-alanine or D-alanine. Fluorescence images are z stacks subjected to deconvolution. Mitochondrial morphology was quantified in 100 cells per sample (triplicate experiments ± SD). **(E and F)** Wild-type and Δ*mip1* cells expressing Su9-DAO were analyzed as in (C and D) (3 biological replicates ± SD). **(G)** Ten-fold serial dilutions of wild-type or Δ*mip1* cells were spotted on agar plates containing minimal complete medium with glucose and incubated at 30˚C. **(H and I)** Wild-type cells expressing either Su9-DAO together with mtGFP (left) or only mtGFP (right) and wild-type cells expressing Su9-DAO together with mtERFP were incubated for 30 min with L-alanine or D-alanine, mixed, and incubated for another 3 h to allow mating and zygote formation. Fluorescence images are z stacks subjected to deconvolution

and were merged with their corresponding DIC image. Medial buds of zygotes are marked with an asterisk. Mitochondrial fusion was quantified in 50 zygotes per sample (triplicate experiments, error bars indicate SD calculated from the sum of zygotes showing complete or partial fusion). Bars, 5 μm. Data pooling and statistics are detailed in S2 Table. The data underlying this figure can be found in S1 Datasheet. DAO, D-amino acid oxidase; DIC, differential interference contrast; mtERFP, mitochondria-targeted enhanced red fluorescent protein; mtGFP, mitochondria-targeted GFP; SCD, synthetic complete dextrose medium; SD, standard deviation; TMRM, tetramethylrhodamine methyl ester.

Approximately 43% of the buds showed strongly reduced mitochondrial inheritance and 37% were completely devoid of mitochondria (Fig 3A and 3B). We conclude that mitochondrially produced $H_2O_2$ severely impairs mitochondrial motility and inheritance.

Next, we tested whether mitochondrially produced $H_2O_2$ affects cytoskeleton-dependent transport processes in general. Fluorescent staining of filamentous actin showed that Su9-DAO-expressing cells maintained normal actin cables and actin patches in D-alanine-containing medium (Fig 3C and 3D). Also the polarized distribution of Myo2-GFP was not affected under these conditions (Fig 3E and 3F) suggesting that Myo2 activity can be maintained. To further test this, we examined the inheritance of two other Myo2 cargo organelles, vacuoles and peroxisomes. The inheritance of vacuoles was impaired to an extent similar to mitochondria (Fig 3G and 3H). Albeit the reason for this phenotype is currently unclear it was not entirely unexpected, since it is known that the inheritance of vacuoles and mitochondria is tightly coordinated [59,60]. In contrast, inheritance of peroxisomes was not affected by $H_2O_2$ production in mitochondria (Fig 3G and 3H) demonstrating that Myo2 is still capable to transport organelles to the bud under these conditions. In sum, these observations suggest that mitochondria-derived $H_2O_2$ impairs mitochondrial inheritance, but does not affect the capacity of Myo2-dependent organelle transport in general.

## Mitochondrial dynamics or retrograde actin cable flow are not critical for partitioning of ROS-stressed mitochondria

We have previously reported that transport of a critical mitochondrial quantity to the bud depends on a fine-tuned balance of anterograde movement powered by Myo2 and mitochondrial fusion and fission. Δfzo1 mutants lacking the major mitochondrial fusion protein, Fzo1, contain fragmented mitochondria and show a significant inheritance defect suggesting that fragmented mitochondria are less efficiently transported into the bud than fused and tubular mitochondria [61]. Therefore, we asked whether the impairment of mitochondrial inheritance under ROS stress is caused by mitochondrial fragmentation. To test this, we quantified mitochondrial inheritance in strains that are unable to undergo mitochondrial fragmentation. Δdnm1 mutants lack the major mitochondrial fission protein, Dnm1, and contain fused, interconnected mitochondrial networks due to a block of mitochondrial fission with ongoing mitochondrial fusion. Δdnm1 Δfzo1 cells have tubular mitochondria, similar to the wild type, which are no longer dynamic due to simultaneous blocks of fusion and fission [62,63]. Mitochondria were efficiently transported into medium-sized buds in Su9-DAO-expressing cells in the presence of L-alanine. Induction of mitochondrial ROS stress by the addition of D-alanine resulted in mitochondrial inheritance defects in fission-defective mutants, very similar to the wild type, while mitochondria maintained an interconnected morphology (Fig 4A and 4B). Also, Su9-DAO-dependent growth defects were similar in Δdnm1 Δfzo1 double and Δdnm1 single mutant and wild-type strains (Fig 4C). We conclude that impaired inheritance of ROS-stressed mitochondria is not caused by mitochondrial fragmentation.

Bundles of actin filaments (cables) serve as tracks for Myo2-dependent organelle transport towards the bud. Actin polymerization in the bud or bud neck and pulling forces produced by myosins generate a constant flow of actin cables towards the pole of the mother cell that is

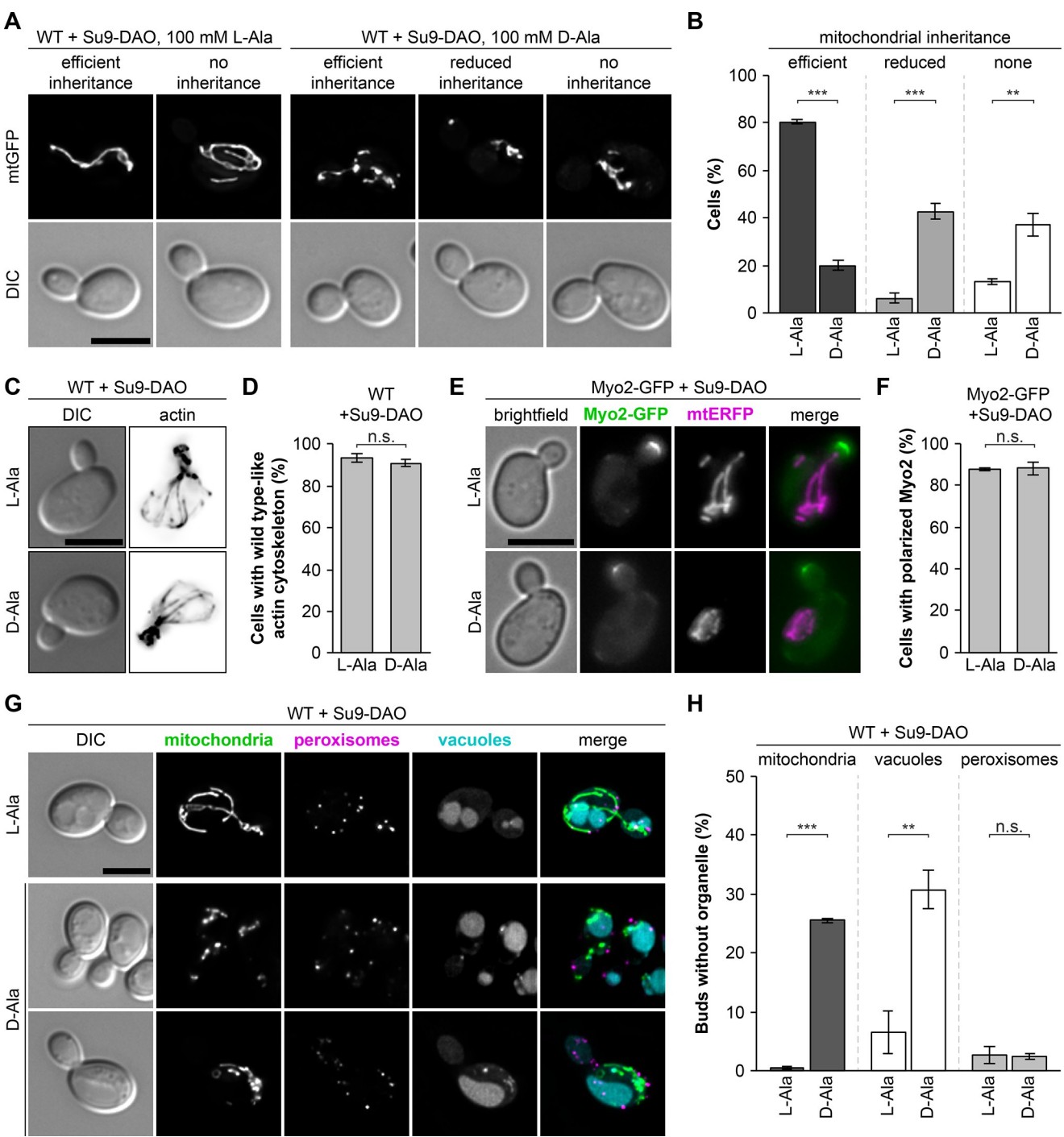

**Fig 3. Mitochondrially produced H₂O₂ impairs mitochondrial motility and inheritance.** (A and B) Wild-type cells expressing Su9-DAO and mtGFP were incubated for 3 h in the presence of L-alanine or D-alanine. Fluorescence images are z stacks subjected to deconvolution. Mitochondrial inheritance was quantified in at least 100 medium-sized buds per sample (triplicate experiments ± SD). **(C and D)** Cells were grown as in (A) and the actin cytoskeleton was stained with rhodamine phalloidine. Fluorescence images are z stacks subjected to deconvolution. Actin morphology was quantified in 100 cells per sample (triplicate experiments ± SD). **(E and F)** Cells expressing Myo2-GFP and mtERFP were grown as in (A). Fluorescence images are maximum intensity projections of z stacks. Myo2-GFP localization was quantified in at least 100 cells per sample (triplicate experiments ± SD). **(G and H)** Wild-type cells expressing mtGFP (green), DsRed-PTS1 (magenta), and Su9-DAO were incubated for 4 h in the presence of L-alanine or D-alanine. The vacuole was stained with CMAC (cyan). Fluorescence images are z stacks subjected to deconvolution. Organellar inheritance was quantified in at least 100 medium-sized buds per sample (triplicate experiments ± SD). Bars, 5 μm. Data pooling and statistics are detailed in S2 Table. The data underlying this figure can be found in S1 Datasheet. DAO, D-amino acid oxidase; mtERFP, mitochondria-targeted enhanced red fluorescent protein; mtGFP, mitochondria-targeted GFP; SD, standard deviation.

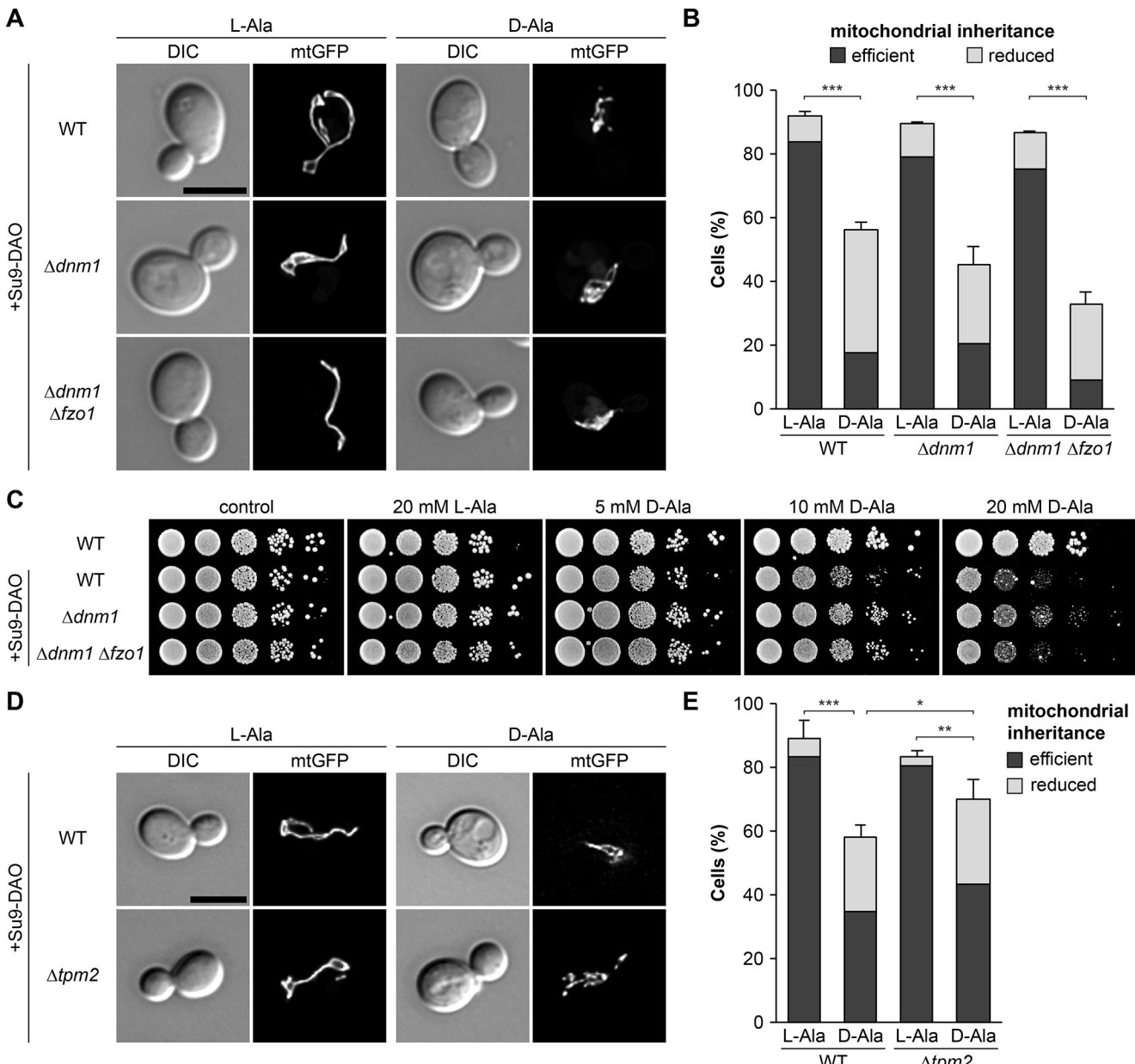

**Fig 4. Retention of ROS-stressed mitochondria is independent of mitochondrial dynamics and actin cable dynamics. (A and B)** Cells expressing Su9-DAO and mtGFP were incubated for 3 h in the presence of L-alanine or D-alanine. Fluorescence images are z stacks subjected to deconvolution. Mitochondrial inheritance was quantified in 100 cells per sample (triplicate experiments, error bars indicate SD calculated from the sum of cells showing reduced or efficient inheritance). In this and the following figures, the percentage of cells lacking mitochondria in their bud corresponds to the difference between the sum of the categories "efficient" plus "reduced" and 100%. **(C)** Ten-fold serial dilutions of cells were spotted on agar plates containing minimal complete medium with glucose and incubated at 30°C. **(D and E)** Cells were analyzed as in (A and B). Bars, 5 μm. Data pooling and statistics are detailed in S2 Table. The data underlying this figure can be found in S1 Datasheet. DAO, D-amino acid oxidase; mtGFP, mitochondria-targeted GFP; SD, standard deviation.

distal to the bud. This retrograde actin cable flow (RACF) constitutes an opposing force that has to be overcome by mitochondria to reach the bud [24,27]. RACF is promoted by the class II myosin, Myo1, and limited by the tropomyosin, Tpm2 [64]. It has been reported that deletion of the *TPM2* gene enhances the asymmetry of mitochondrial redox state in the mother

cell and bud, whereas deletion of *MYO1* abrogates it. Based on these observations, it was suggested that increased RACF in Δ*tpm2* selects more reduced mitochondria for inheritance, whereas reduced RACF in Δ*myo1* allows the entry of more oxidized mitochondria into the bud. Thus, RACF might serve as a filter to prevent low-functioning (i.e., more oxidized) mitochondria from moving into the bud [24].

To test whether RACF affects the inheritance of mitochondria containing active Su9-DAO, we quantified mitochondria in buds of Su9-DAO-expressing Δ*tpm2* cells. If RACF plays a major role in preventing these mitochondria from entering the bud, it can be expected that Δ*tpm2* buds contain fewer mitochondria than the wild type. However, we observed that their mitochondrial content was even slightly increased (Fig 4D and 4E), suggesting that increased RACF is not sufficient to prevent dysfunctional mitochondria from moving into the bud.

Deletion of the *MYO1* gene is lethal in some genetic backgrounds [65,66] and produces severe cytokinesis defects in others [67–69]. We generated viable haploid Δ*myo1* mutants by tetrad dissection of a heterozygous diploid deletion mutant and observed that cells were mis-shapen and/or multibudded, and buds were very heterogeneous in size (S3 Fig). Thus, it was not possible to analyze the role of Myo1 in mitochondrial inheritance in the genetic background used in this study.

## Inheritance of ROS-stressed mitochondria depends on recruitment of Myo2

We reasoned that bud-directed transport under ROS stress may be restored by reinforced recruitment of the mitochondrial Myo2 motor. Overexpression of either of the two factors that promote binding of Myo2 to mitochondria, Mmr1 and Ypt11, has been shown to enhance mitochondrial transport to the bud [33–35,61]. Strikingly, the overexpression of Mmr1 or Ypt11 restored mitochondrial inheritance under ROS stress to a level similar to non-stressed conditions (Fig 5A–5D). In contrast, inheritance of mitochondria in cells overexpressing the *ypt11(G40D)* allele, which encodes an inactive Ypt11 variant with a mutation in its GTPase domain [29], remained ROS stress-sensitive (Fig 5A and 5B). These observations suggest that enhanced motor recruitment restores bud-directed mitochondrial transport in the presence of mitochondrially produced $H_2O_2$.

To further test this, we examined mitochondrial inheritance in cells expressing Myo2-Fis1. In this construct, the C-terminal cargo binding domain of Myo2 is replaced by the membrane anchor of the mitochondrial outer membrane protein Fis1 [31]. Thus, Myo2-Fis1 is permanently inserted in the mitochondrial outer membrane and therefore bypasses the functions of Mmr1 and Ypt11 in motor recruitment [31,33]. Strikingly, Myo2-Fis1-mediated transport of mitochondria into the bud was insensitive to mitochondrially produced $H_2O_2$ (Fig 5E and 5F). In sum, these data suggest that enforced recruitment of Myo2 promotes transport of ROS-stressed mitochondria. In other words, reduced recruitment of Myo2 in the presence of mitochondrially produced $H_2O_2$ might be responsible for a lack of mitochondrial movements under these conditions.

## Intracellular distribution of Ypt11 and Mmr1 is affected by mitochondrial ROS stress

As recruitment of Myo2 appears to be critical for asymmetric inheritance of mitochondria under ROS stress, we further examined the roles of Ypt11 and Mmr1. GFP-Ypt11 expressed from plasmids at slightly elevated levels rescues mitochondrial inheritance defects in Δ*ypt11* and localizes to bud tips, thereby mirroring the localization of Myo2 [32]. Consistently, GFP-Ypt11 colocalized with mitochondria in tips of small or medium-sized buds (Fig 6A).

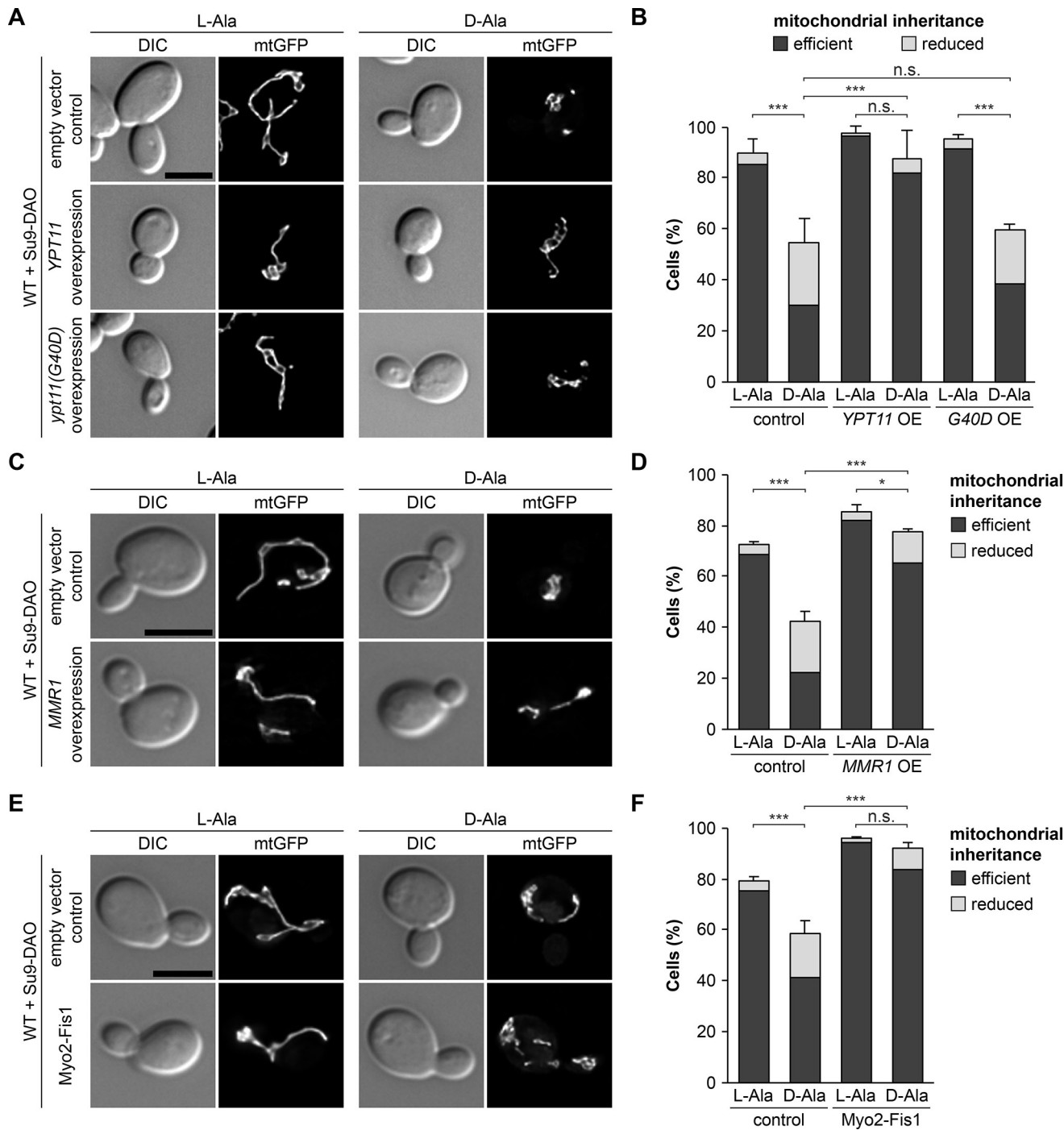

**Fig 5. Enhanced recruitment of Myo2 promotes transport of ROS-stressed mitochondria into the bud. (A–F)** Cells expressing Su9-DAO and mtGFP were incubated for 3 h in the presence of L-alanine or D-alanine. Ypt11 and Ypt11(G40D) were overexpressed from plasmids pAG415GPD-YPT11 and pAG415GPD-ypt11(G40D), Mmr1 was overexpressed from plasmid pAG415GPD-MMR1, and Myo2-Fis1 was expressed from plasmid pRS416-myo2-fis1. Fluorescence images are z stacks subjected to deconvolution. Mitochondrial inheritance was quantified in at least 100 cells per sample (triplicate experiments, error bars indicate SD calculated from the sum of cells showing reduced or efficient inheritance). Bars, 5 μm. Data pooling and statistics are detailed in S2 Table. The data underlying this figure can be found in S1 Datasheet. DAO, D-amino acid oxidase; mtGFP, mitochondria-targeted GFP; ROS, reactive oxygen species; SD, standard deviation.

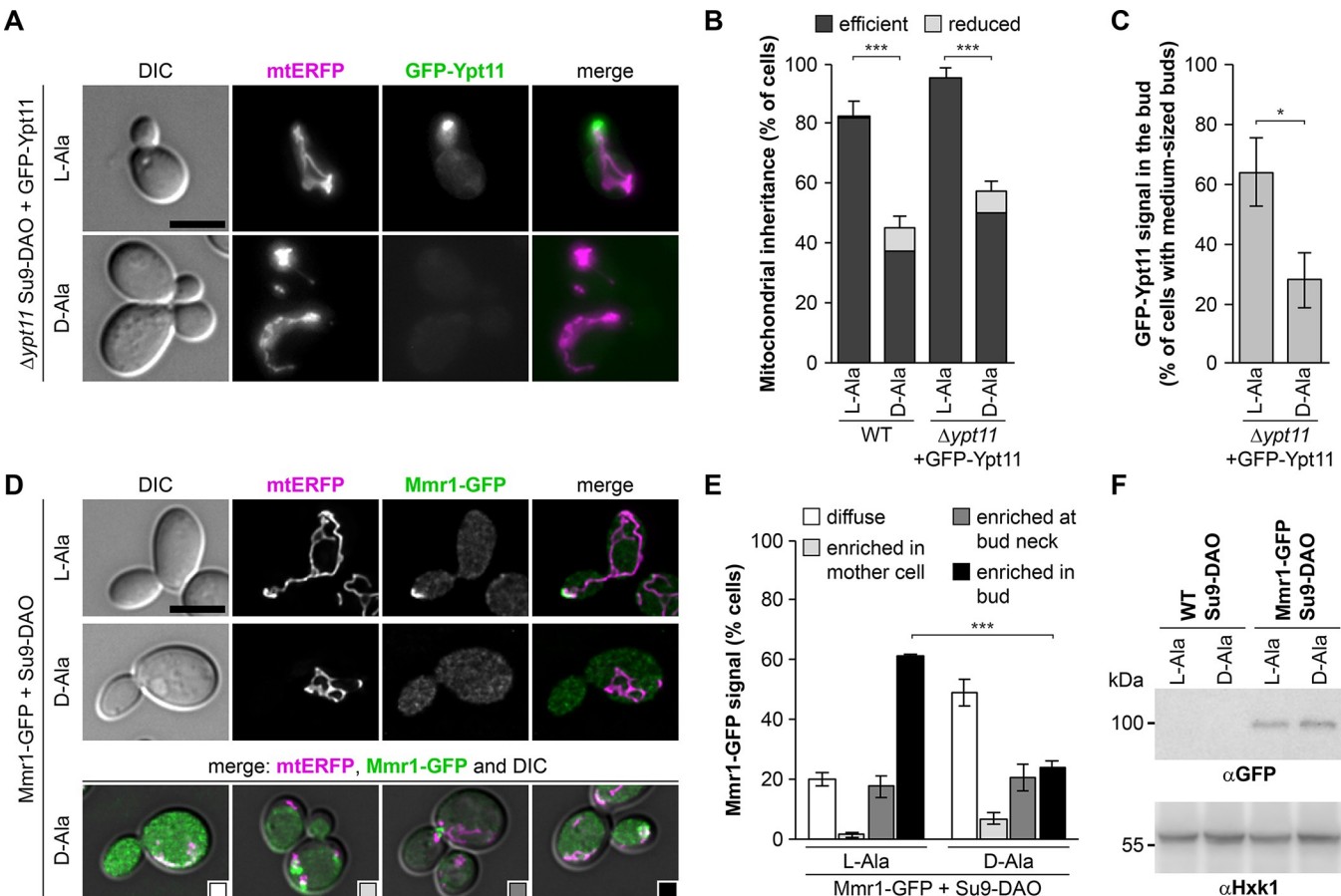

**Fig 6. Localization of Ypt11 and Mmr1 is altered upon mitochondrial ROS stress. (A and B)** Δ*ypt11* cells expressing mtERFP and Su9-DAO were transformed with plasmid p416 (MET25)-GFP-YPT11, grown in selective medium, and analyzed by fluorescence microscopy using identical camera settings. Fluorescence images are maximum intensity projections of z stacks. Mitochondrial inheritance was quantified in 100 cells per sample (triplicate experiments, error bars indicate SD calculated from the sum of cells showing reduced or efficient inheritance). **(C)** Cells were grown as in (A) and medium-sized buds lacking a GFP-Ypt11 signal were quantified in 100 cells per sample (triplicate experiments ± SD). **(D and E)** Cells carrying a genome-inserted *MMR1-yEFGP* allele and expressing mtERFP and Su9-DAO were analyzed by fluorescence microscopy. Fluorescence images are z stacks subjected to deconvolution. Cells were grouped into four categories: (i) diffuse Mmr1-GFP signal, (ii) signal enriched in the mother cell, (iii) at the bud neck, or (iv) in the bud. Representative cells for each category are shown in the bottom panel. Mmr1-GFP was quantified in 100 cells per sample (triplicate experiments ± SD). Bars, 5 μm. Data pooling and statistics are detailed in S2 Table. The data underlying this figure can be found in S1 Datasheet. **(F)** *MMR1* (WT) and *MMR1-yEFGP* (Mmr1-GFP) cells expressing Su9-DAO were grown in the presence of L-alanine or D-alanine and cell extracts were analyzed by Western blotting using anti-GFP antibodies. Hexokinase (Hxk1) served as a loading control. DAO, D-amino acid oxidase; mtERFP, mitochondria-targeted enhanced red fluorescent protein; ROS, reactive oxygen species; SD, standard deviation.

Mitochondrial inheritance was still ROS stress-sensitive in GFP-Ypt11 expressing cells (Fig 6B), and mitochondrial ROS stress led to the disappearance of GFP-Ypt11 from buds (Fig 6A and 6C), likely reflecting the mitochondrial inheritance defect under these conditions.

Next, we tagged the *MMR1* gene with *yEGFP* in the chromosome. The *MMR1-yEGFP* allele is functional as cells containing this allele together with the Δ*ypt11* allele are viable (S4 Fig). Mmr1-GFP colocalized with mitochondria in tips of small and medium-sized buds or at the bud neck of large budded cells, again mirroring the localization of Myo2. This polarized distribution of Mmr1-GFP was largely lost upon induction of $H_2O_2$ production in mitochondria (Fig 6D and 6E). The loss of focal Mmr1-GFP signals in the majority of stressed cells suggests that foci containing active Mmr1/Myo2 complexes disappear from the surface of $H_2O_2$ producing mitochondria.

Recently, it was shown that Mmr1 is subject to ubiquitination and degradation when newly inherited mitochondria reach the bud tip [70]. We asked whether enhanced turnover of Mmr1 can be triggered by mitochondrially produced $H_2O_2$. To test this, we determined Mmr1-GFP levels in Western blots of total cell extracts of Su9-DAO expressing cells. We observed that Mmr1-GFP levels were very similar in cells grown in L-alanine and D-alanine-containing media (Fig 6F). This suggests that Mmr1 is released from stressed and immobilized mitochondria, but not subsequently degraded.

## Mmr1 is critical for retention of ROS-stressed mitochondria in mother cells

If Ypt11 and/or Mmr1 play a critical role in retention of mitochondria upon ROS stress, mitochondrial inheritance should be rendered ROS-insensitive when the *YPT11* or *MMR1* gene is deleted. Therefore, we examined mitochondrial inheritance in Δ*ypt11* or Δ*mmr1* mutant cells in the absence or presence of Su9-DAO activity. Consistent with published observations [29,34], Δ*ypt11* and Δ*mmr1* single mutants showed a moderate mitochondrial inheritance defect under non-stressed conditions (Fig 7A and 7B). Production of $H_2O_2$ in mitochondria resulted in a severe reduction of mitochondrial inheritance in wild-type and Δ*ypt11* cells. Strikingly, we found that deletion of *MMR1* rendered mitochondrial inheritance insensitive to mitochondrial ROS stress, i.e., mitochondrial inheritance in Su9-DAO expressing Δ*mmr1* cells was very similar in L-alanine and D-alanine-containing media (Fig 7B). This suggests that Mmr1 is critical for the retention of ROS-stressed mitochondria in yeast mother cells.

To further test this idea, we constructed a yeast strain that is able to recruit Myo2 to mitochondria independent of Mmr1 and Ypt11. Inp2 is the Myo2 receptor on peroxisomes and interacts with the subdomain of the Myo2 cargo-binding domain distal of the Mmr1 binding site [59,71,72]. We fused the large, cytosol-exposed part of Inp2 to the membrane anchor of the mitochondrial outer membrane protein Tom20 and GFP. Tom20-Inp2-GFP formed punctate structures that largely colocalized with mitochondria in yeast cells (Fig 7C). Importantly, expression of this construct rescued lethality of the Δ*mmr1* Δ*ypt11* double deletion, indicating that it replaces the function of the mitochondrial Myo2 adaptor proteins (Fig 7D). Mitochondria were inherited as efficiently as in wild-type cells under non-stressed conditions. Remarkably, their inheritance was not affected by mitochondrially generated $H_2O_2$ (Fig 7E and 7F). Thus, replacement of Mmr1/Ypt11 by Tom20-Inp2-GFP renders mitochondrial inheritance insensitive to ROS stress. We conclude that the presence of Mmr1 is critical for retention of damaged mitochondria in mother cells.

## Retention of ROS-stressed mitochondria in mother cells is a highly selective process

So far, our results suggest that oxidative damage results in immobilization of mitochondria triggered by the release or inactivation of Myo2 and Mmr1. However, it is unclear whether this is a general stress response that affects all mitochondria in the cell. Alternatively, the cellular transport machinery might be able to differentiate between stressed and non-stressed mitochondria to selectively transport healthy mitochondria and thereby retain damaged organelles in mother cells. To test this, we developed an assay that allowed us to simultaneously observe the behavior of individual mitochondria that share the same cytosol in the same cell.

To be able to track the behavior of individual mitochondria over a long period of time, we wanted to avoid fusion and matrix content mixing. Therefore, we employed the Δ*dnm1* Δ*fzo1* double mutant, whose mitochondria cannot fuse but have a wild type-like morphology [62,63] (see also Fig 4A). We constructed a Δ*dnm1* Δ*fzo1* strain that is *MAT*a and expresses Su9-DAO

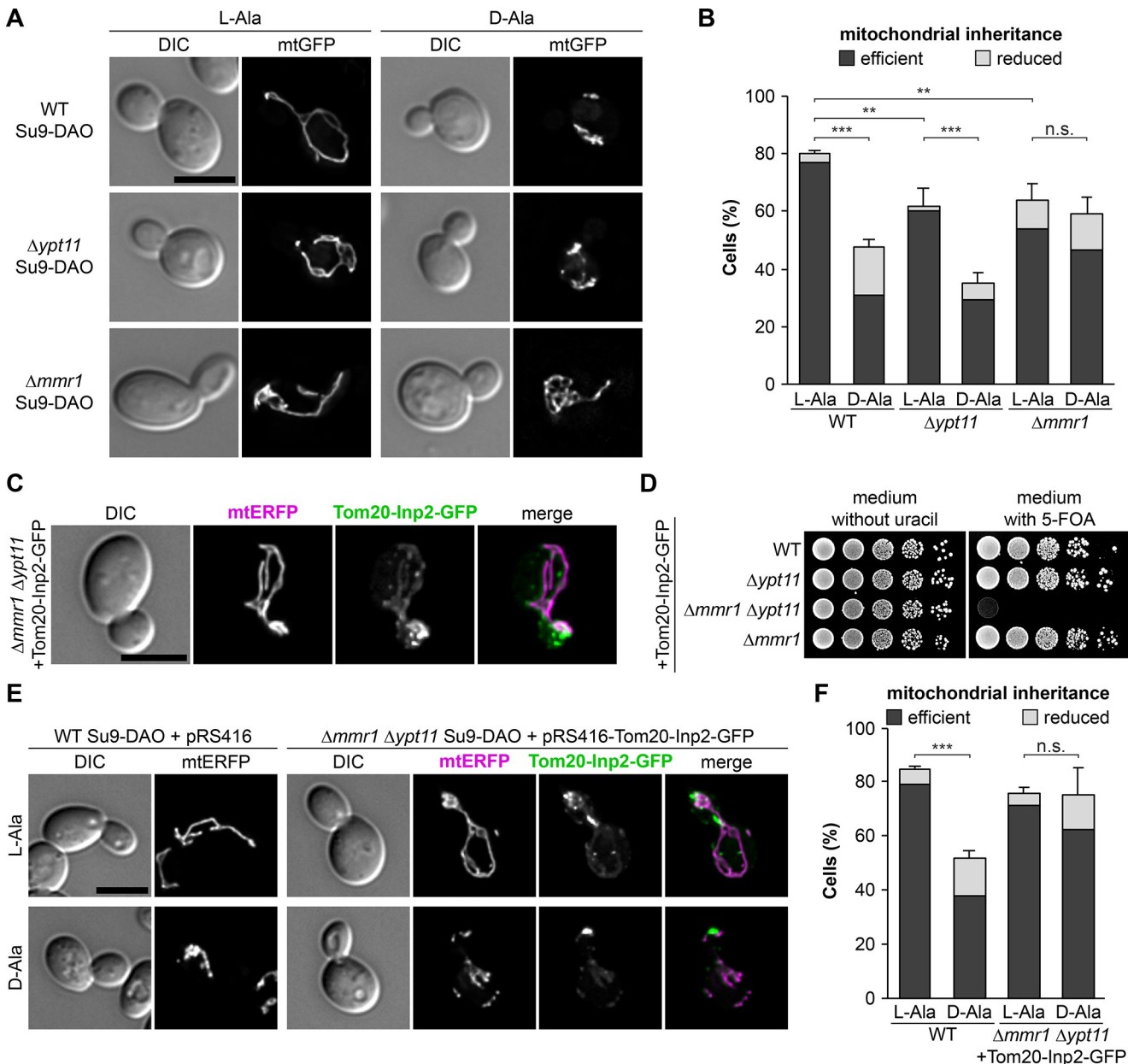

**Fig 7. Mmr1 is required for retention of damaged mitochondria in mother cells. (A and B)** Cells expressing Su9-DAO and mtGFP were incubated for 3 h in the presence of L-alanine or D-alanine. Fluorescence images are z stacks subjected to deconvolution. Mitochondrial inheritance was quantified in at least 100 cells per sample (triplicate experiments, error bars indicate SD calculated from the sum of cells showing reduced or efficient inheritance). **(C)** Δmmr1 Δypt11 cells expressing Tom20-Inp2-GFP and mtERFP were grown to logarithmic growth phase and analyzed by fluorescence microscopy. Fluorescence images are z stacks subjected to deconvolution. **(D)** Wild-type, single, and double mutant strains containing a *URA3*-based plasmid encoding Tom20-Inp2-GFP were allowed to lose the plasmid in medium containing uracil. Then, 10-fold serial dilutions of cells were spotted on agar plates containing minimal complete medium either without uracil or supplemented with uracil and 5-FOA, which counterselects against the *URA3* marker. Lack of growth on 5-FOA medium indicates inability to lose the plasmid; i.e., the gene encoded by the plasmid is essential in this background. **(E and F)** Wild-type cells expressing Su9-DAO and mtERFP and containing plasmid pRS416 (empty vector control) and Δmmr1 Δypt11 cells expressing Su9-DAO, Tom20-Inp2-GFP, and mtERFP were incubated for 3 h in the presence of L-alanine or D-alanine. Fluorescence images are z stacks subjected to deconvolution. Mitochondrial inheritance was quantified in 100 cells per sample (triplicate experiments, error bars indicate SD calculated from the sum of cells showing reduced or efficient inheritance). Bars, 5 μm. Data pooling and statistics are detailed in S2 Table. The data underlying this figure can be found in S1 Datasheet. 5-FOA, 5-fluoroorotic acid; DAO, D-amino acid oxidase; mtERFP, mitochondria-targeted enhanced red fluorescent protein; mtGFP, mitochondria-targeted GFP; SD, standard deviation.

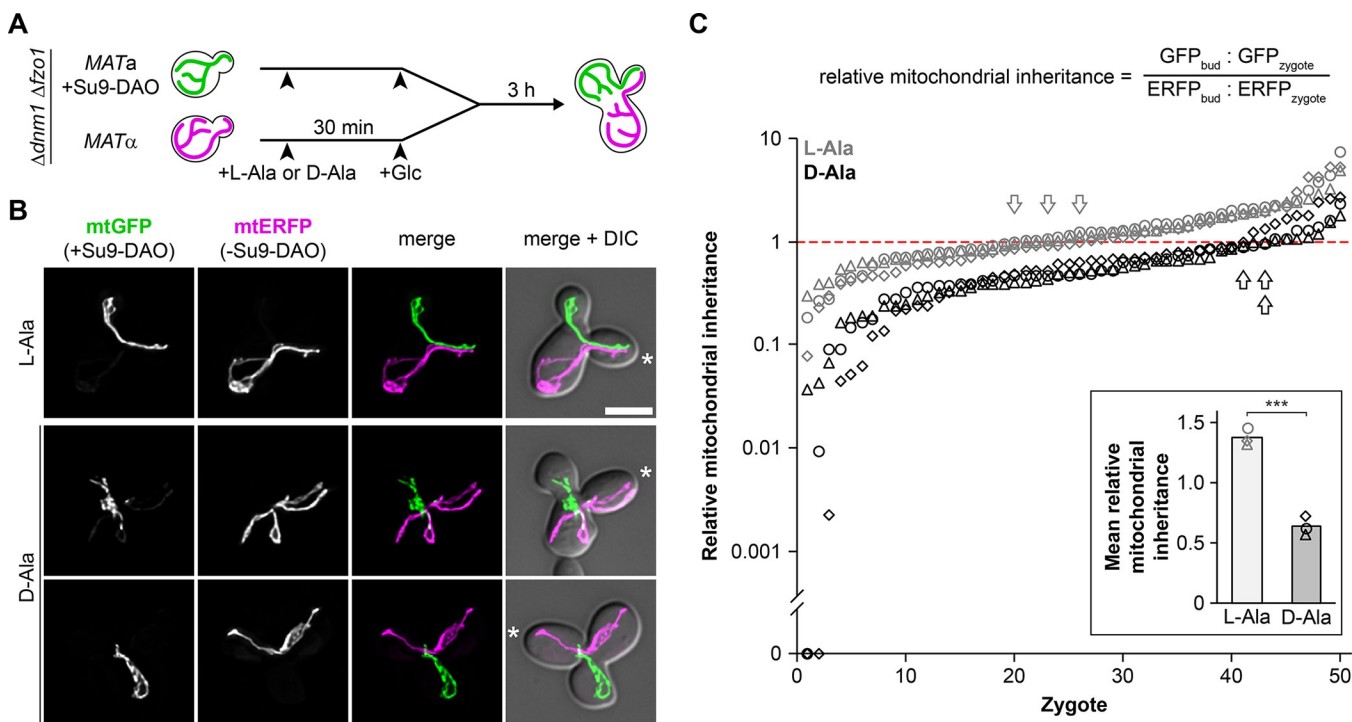

**Fig 8. Healthy mitochondria are selectively inherited in buds formed at zygotes. (A)** Schematic outline of the experiment: Δ*dnm1* Δ*fzo1* cells expressing Su9-DAO and mtGFP (green) or only mtERFP (magenta) were grown in L-alanine (control) or D-alanine (ROS stress)-containing media, expression of fluorescent proteins was stopped by addition of glucose to the media, cells were mated and allowed to form buds for 3 h. **(B)** Fluorescence images are z stacks subjected to deconvolution. Asterisks indicate medial buds of zygotes. Bar, 5 μm. **(C)** Relative mitochondrial inheritance was quantified in 50 zygotes per sample by calculating the GFP and ERFP intensity ratios as shown in the equation. Arrows point to the zygote that is closest to the value of 1 for each experiment (gray, L-alanine; black, D-alanine). Inset, mean values of relative mitochondrial inheritance (arbitrary units; triplicate experiments). Data pooling and statistics are detailed in S2 Table. The data underlying this figure can be found in S1 Datasheet. DAO, D-amino acid oxidase; mtERFP, mitochondria-targeted enhanced red fluorescent protein; mtGFP, mitochondria-targeted GFP; ROS, reactive oxygen species.

and mtGFP, and another Δ*dnm1* Δ*fzo1* strain that is *MAT*α and expresses mtERFP but lacks the Su9-DAO construct. Cells were pre-cultured in medium containing galactose to induce expression of the fluorescent mitochondrial marker proteins and either L-alanine (i.e., non-stressed control) or D-alanine (i.e., induction of mitochondrial ROS stress). Then, glucose was added to shut off the synthesis of mtGFP and mtERFP, and cultures were mixed to allow mating and zygote formation. Cells were then further incubated to allow bud formation at the zygote, and inheritance of differently labeled mitochondria was quantified (Fig 8A).

We observed that both mating partners passed on their mitochondria equally efficiently to the newly formed bud in the presence of L-alanine. When mitochondrial ROS stress was selectively induced in one mating partner by the addition of D-alanine, we observed a strong preference for transmission of mitochondria from the non-stressed counterpart. Intriguingly, more than 80% of the buds received mitochondria mainly or exclusively from the mating partner that lacked the Su9-DAO construct (Fig 8B and 8C). To exclude that this was due to the mating type or other strain-specific effects, we constructed a new set of strains with switched mating types and obtained the same result (S5 Fig). Analysis of zygotes by live cell microscopy showed that mitochondria contributed by the stressed parental cell were less mobile in the zygote, whereas mitochondria contributed by the non-stressed parental cell migrated efficiently into the emerging bud (S6 and S7 Videos).

These observations demonstrate that mitochondria that are present in the same cell behave remarkably differently depending on their physiological conditions. DAO activity in the matrix renders them almost immotile and prevents their entry into the newly formed bud, whereas mitochondria without DAO activity continue to show efficient movements directed towards the bud. Obviously, these strikingly different behaviors are not due to a general deterioration of cell physiology or inactivation of cytosolic factors or cytoskeletal tracks. Rather, it appears that mitochondrial motility is regulated directly at the organellar level.

## Mmr1 is critical for selective inheritance of healthy mitochondria

Finally, we analyzed the selective transmission of healthy mitochondria and retention of dysfunctional mitochondria in mitotically dividing cells. We placed Su9-DAO under control of the repressible *GAL* promoter and modified the zygote assay as follows. Δ*dnm1* Δ*fzo1* strains of opposite mating types with and without $P_{GAL}$-*Su9-DAO* were grown in galactose-containing medium to pre-load mitochondria with either GFP and DAO or with ERFP only. Glucose was added to stop expression of these proteins, cultures were mixed to allow mating, and L-alanine or D-alanine was added 3 h after zygote formation. This regime allowed the formation of diploid cells containing ROS-stressed (green) together with non-stressed (magenta) mitochondria. These cells were incubated until they formed buds, and mitochondrial inheritance was quantified in buds of diploid mothers that contained both mitochondrial populations (Fig 9A).

This assay nicely recapitulated the results obtained with the previous zygote assay (compare Fig 8). In the absence of ROS stress more than 80% of progeny budding off from diploid cells received mitochondria derived from both parents. In contrast, the transmission of dysfunctional mitochondria was efficiently blocked upon the addition of D-alanine and about 70% of buds contained exclusively non-stressed mitochondria (Fig 9B and 9C and S8 and S9 Videos). We observed that ROS-stressed mitochondria that remain in the mother were often located very close to non-stressed mitochondria moving into the daughter (Fig 9B and S9 Video). As these mitochondria nevertheless behaved differently, we suggest that their behavior is not determined by $H_2O_2$ or other soluble factors leaking out of the organelle and eliciting a cytosolic response, but rather by signals on the organelle itself.

To analyze whether Ypt11 and/or Mmr1 are required for selective inheritance of healthy mitochondria, we tested Δ*dnm1* Δ*fzo1* Δ*ypt11* and Δ*dnm1* Δ*fzo1* Δ*mmr1* triple mutants. Deletion of the *YPT11* gene did not affect selective mitochondrial inheritance (Fig 9B and 9C). Astonishingly, the deletion of the *MMR1* gene completely abolished the selection of non-stressed mitochondria. Even under control conditions, the inheritance pattern appeared to be more disordered, and relatively many buds contained mitochondria derived from only one parent. This pattern did not change at all upon induction of Su9-DAO activity (Fig 9B and 9C), indicating that cells lacking Mmr1 have lost the ability to discriminate between stressed and non-stressed mitochondria. We conclude that the presence of Mmr1, but not Ypt11, is crucial for the selective and ordered transport of healthy mitochondria into the bud and thus retention of damaged mitochondria in mother cells.

## Discussion

It has been previously hypothesized that oxidatively damaged mitochondria are retained in mother cells and preferential inheritance of fit mitochondria contributes to rejuvenation of daughter cells [13,73,74]. However, experimental evidence was rather limited or indirect. Mitochondria were found to be highly fragmented and have declining membrane potential in aged mother cells [21,60]; active aconitase was preferentially found in buds when yeast cells

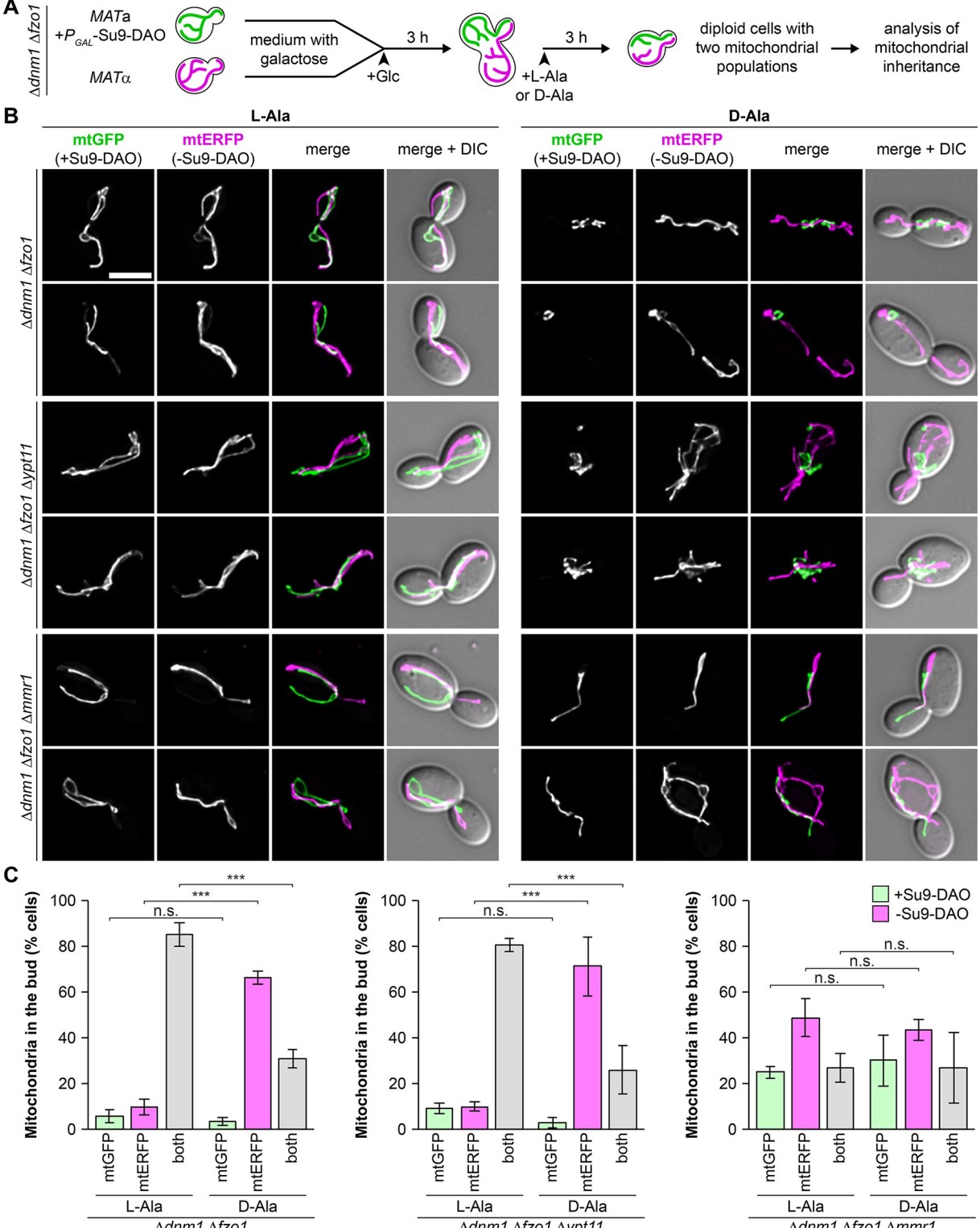

**Fig 9. Mmr1 is critical for selective inheritance of healthy mitochondria during mitotic growth. (A)** Schematic outline of the experiment: Δdnm1 Δfzo1 cells were preloaded with Su9-DAO and mtGFP (green) or only mtERFP (magenta), expression was stopped by addition of glucose to the medium, cells were mated, allowed to form buds in the presence of L-alanine (control) or D-alanine (ROS stress) and further incubated for mitotic growth. **(B and C)** Δdnm1 Δfzo1, Δdnm1 Δfzo1 Δypt11, and Δdnm1 Δfzo1 Δmmr1 cells were analyzed as in (A). Fluorescence images are z stacks subjected to deconvolution. Mitochondrial inheritance was quantified in at least 50 cells per sample; only

diploid mother cells carrying both labels were taken into account (at least triplicate experiments ± SD). Bar, 5 μm. Data pooling and statistics are detailed in S2 Table. The data underlying this figure can be found in S1 Datasheet. DAO, D-amino acid oxidase; mtERFP, mitochondria-targeted enhanced red fluorescent protein; mtGFP, mitochondria-targeted GFP; ROS, reactive oxygen species; SD, standard deviation.

have been exposed to external oxidative stress [22], and fluorescent biosensors revealed a small increase in the level of mitochondrial ROS in mother cells compared to buds [23]. Yet, several important questions remained unanswered. Is the unequal distribution of mitochondria a consequence of optimized physiological conditions in the bud post-inheritance, or is it due to the specific selection of healthy mitochondria pre-inheritance? Is there a general decline of mitochondrial motility in cells harboring dysfunctional mitochondria, or can the mother cell specifically select individual mitochondria for transport? What are the components of the cellular transport machinery that are involved?

The aim of our work was to find answers to these questions. We show that mitochondrial ROS stress selectively impairs bud-directed transport of dysfunctional mitochondria. Damaged mitochondria undergo fragmentation, are fusion-incompetent, become immotile, and therefore are retained in the mother cell. Immobilization is specific to the damaged organelle, as unaffected mitochondria continue to move into the bud. This is evident from zygotes or their progeny containing differentially labeled mitochondria with and without Su9-DAO. Quality control and selection for inheritance thus take place at the level of individual organelles in the mother cell. This selection allows the cell to actively separate damaged from intact mitochondria during cell division.

We identified Mmr1 as a key factor that separates healthy mitochondria destined for bud-directed transport from damaged mitochondria that are retained in the mother cell. Our data demonstrate that inheritance of mitochondria is no longer sensitive to ROS stress in cells lacking Mmr1. A role of Mmr1 in mitochondrial quality control during inheritance fits to previously reported observations. Δ*mmr1* cells have a very heterogeneous replicative lifespan; i.e., long-lived Δ*mmr1* cells grow faster, have a more reducing mitochondrial redox potential, and lower ROS level, whereas short-lived Δ*mmr1* cells have the opposite phenotype [23]. Also, *mmr1-S414A* cells carrying a mutation in a phosphorylation motif have elevated ROS levels [70]. Furthermore, overexpression of *MMR1* results in a decline of replicative lifespan [75]. These observations are consistent with the idea that the separation of fit from dysfunctional mitochondria no longer works in *mmr1* mutants or cells that express excess Mmr1, so that some young mothers start with plenty of healthy organelles and others with an accumulation of damaged organelles.

Our results further suggest that Mmr1 functions by regulating the recruitment of Myo2 to mitochondria, since quality control during inheritance no longer works when Mmr1 function is bypassed, e.g., by expression of Myo2-Fis1 or Tom20-Inp2-GFP. Consistently, induction of ROS stress in mitochondria leads to loss of mitochondria-associated Mmr1-GFP. We propose that Mmr1-dependent motor recruitment is the major mechanism to ensure that healthy mitochondria are transported to the bud. Whether RACF provides an additional filtering mechanism to ensure retention of dysfunctional mitochondria [24] is currently unclear, at least from our perspective. Also, our data clearly show that Mmr1-dependent selection of healthy mitochondria destined for bud-directed transport occurs in the mother cell. Therefore, we consider it unlikely that mitochondrial quality control relies on a function of Mmr1 as an anchor at the bud cell cortex, as was previously hypothesized [75].

Mmr1 is a peripheral mitochondrial outer membrane protein that directly interacts with phospholipids [76]. It is a phosphoprotein and a target of the serine/threonine phosphatase Ptc1 [77], which has a known role in mitochondrial inheritance [78]. Interestingly, deletion of

the *PTC1* gene results in mislocalization of Mmr1 to punctate structures throughout mother cells and buds [77]. Recently, it was shown that protein kinases Cla4 and Ste20 phosphorylate Mmr1 in the growing bud, which activates Mmr1 poly-ubiquitination by ubiquitin ligases Dma1 and Dma2. This triggers degradation of Mmr1 to release mitochondria from the actin cytoskeleton at the bud tip [70]. It will be an exciting question for future studies to elucidate the signal that triggers the release of Mmr1 and Myo2 from mitochondria upon ROS stress and to investigate whether Ptc1, Cla4, Ste20, Dma1, Dma2, and/or other factors are involved.

Surprisingly, we observed that both overexpression and deletion of *MMR1* renders mitochondrial inheritance insensitive to Su9-DAO-induced ROS stress. To explain this apparent paradox, we propose that Mmr1 plays an active role as a molecular switch regulating Myo2 recruitment. Therefore, its absence in Δ*mmr1* leads to loss of selectivity in mitochondrial inheritance. As Mmr1 itself is subject to posttranslational modification and regulation of its activity [70,77], it is conceivable that a factor required for its inactivation may become limiting in the presence of excess Mmr1, which would lead to continued inheritance of damaged mitochondria. Alternatively, we consider it likely that some residual activity of Mmr1 remains even when most of the protein has been inactivated under stressed conditions. In the presence of excess Mmr1, this activity might be sufficient to promote Myo2 recruitment and drive transport of dysfunctional mitochondria into the bud. These scenarios are not mutually exclusive and point to a fine-tuned network of regulatory mechanisms that govern selective mitochondrial inheritance.

In sum, we propose the following scenario for mitochondrial quality control during inheritance. When ROS stress or other forms of non-physiological insults cause a breakdown of mitochondrial membrane potential, these mitochondria become fusion incompetent, as inner membrane fusion requires the presence of a membrane potential [79]. This leads to mitochondrial fragmentation by ongoing mitochondrial fission and at the same time prevents fusion and intermixing of damaged with healthy mitochondria. Mitochondrial dysfunction and/or ROS stress is sensed by Mmr1 that sheds off Myo2 from the mitochondrial surface. These mitochondria are thereby immobilized and remain in the mother cell. In contrast, healthy mitochondria are efficiently transported into the bud by Myo2. Poly-ubiquitylation and degradation of Mmr1 at the bud tip then releases Myo2 and thereby prevents transport of newly inherited mitochondria back to the bud neck after the actin cytoskeleton has re-polarized in large buds. Thus, the fine-tuned activities of mitochondrial fission, fusion, and transport ensure the inheritance of the fittest mitochondria to the next generation of cells.

## Materials and methods

### Plasmids

Plasmids pYES-mtGFP, pYX122-mtGFP, pYX142-mtGFP [80], pYX142-mtyEGFP (see below), pYES-mtERFP [61], and pYX142-mtERFP [81] were used to label mitochondria, and plasmid pDsRed-PTS1 [82] was used to label peroxisomes. Myo2-Fis1 was expressed from plasmid pRS416-myo2-fis1 [31]. Plasmids pAG415GPD-YPT11 and pAG415GPD-ypt11 (G40D) were used for overexpression of Ypt11 [61], and plasmid p416(MET25)-GFP-YPT11 was used for low-level expression of GFP-Ypt11 [32].

To construct pYX142-mtyEGFP, the coding sequence of yeast-adapted enhanced GFP was amplified from plasmid pKT209 [83] using primers 5′ TAT ATA AGA TCT ATG TCT AAA GGT GAA GAA TTA and 5′ TAT ATA CTC GAG CTG CCG GTA GAG GTG TGG TC and cloned into the *Bgl*II and *Xho*I sites of pYX142-mtGFP [80].

To construct plasmid pAG415GPD-MMR1 used for overexpression of Mmr1, the *MMR1* ORF was amplified from yeast DNA using primers 5′ GGG GAC AAG TTT GTA CAA AAA

AGC AGG CTT CAT GAA TTC TCC AAC AAT GAA ATC G and 5′ GGG GAC CAC TTT GTA CAA GAA AGC TGG GTC TCA TTT TTC CTT CTT CTC CAC TTG G and inserted into pAG415GPD by Gateway cloning [84].

To construct pXRnX-Su9-DAO, a *Sac*I/*Eag*I restriction fragment containing the *Su9-DAO* coding sequence together with the *TEF* promoter was subcloned from p413-Su9-DAO-FLAG [57] into the integrating vector (lacking ARS/CEN) pXRnX [85]. This plasmid was used to integrate *Su9-DAO* under control of the *TEF* promoter into the *HO* locus of the yeast genome.

Plasmid pXRnX-pGAL-Su9-DAO was generated by overlap extension PCR. The *GAL* promoter was amplified from the pYES2.0 vector (Invitrogen) using primers 5′ TAT ATA GAG CTC AGT ACG GAT TAG AAG CCG CCG AGC GGG TGA CAG C and 5′ CGA GGA CAC GAG TGG AGG CCA TTC TAG AGT TTT TTC TCC TTG ACG TTA AAG adding a *Sac*I restriction site. *Su9-DAO* was amplified from pXRnX-Su9-DAO using primers 5′ CTA TAC TTT AAC GTC AAG GAG AAA AAA CTC TAG AAT GGC CTC CAC TCG TGT CC and 5′ TAA TTA CGG CCG CAA ATT AAA GCC TTC GAG CGT CCC AAA ACC containing the downstream *Eag*I restriction site. Overlap PCR generated a DNA fragment that was cloned into the *Sac*I and *Eag*I sites of the integrating vector (lacking ARS/CEN) pXRnX [85]. This plasmid was used to integrate *Su9-DAO* under control of the *GAL* promoter into the *HO* locus of the yeast genome.

Plasmid pRS416-Tom20-Inp2-GFP was generated by overlap extension PCR. The *TOM20* promoter and the region coding for the N-terminal Tom20 transmembrane segment were amplified using primers 5′ ATA TCT CGA GAT AAC TCT CGT CCA GAA TGC, adding a *Xho*I site, and 5′ AAG GTT TTA TAG CAA ATC AAG CCG CCG CTA TTT CTT CTT TGA TAG T; the region coding for the cytosolic domain of Inp2 was amplified using primers 5′ ACT ATC AAA GAA GAA ATA GCG GCG GCT TGA TTT GCT ATA AAA CCT T and 5′ AAC ACC AGT GAA TAA TTC TTC ACC TTT AGA GCC GCC TGA ATC ATT TCC TAG TAA TC; the region coding for yeast-enhanced GFP (yEGFP) together with an *ADH* terminator was amplified from plasmid pKT209 [83] using primers 5′ GAT TAC TAG GAA ATG ATT CAG GCG GCT CTA AAG GTG AAG AAT TAT TCA CTG GTG TT and 5′ TTA AGG ATC CTG CCG GTA GAG GTG TGG TCA ATA, adding a *Bam*HI site. Overlap PCR generated a DNA fragment that was cloned into the *Xho*I and *Bam*HI sites of pRS416 [86]. The final construct encodes the following chimeric protein: Tom20(1–36)-Gly-Gly-Inp2(240–705)-Gly-Gly-yEGFP.

## Yeast strains

All yeast strains are derivatives of BY4741 and BY4742 [87]. Standard methods were used for strain construction [88,89]. Deletion mutants were either taken form the yeast deletion collection [66] or constructed by PCR-based gene deletion. Haploid Δ*myo1* strains were obtained by sporulation and tetrad dissection of a *MYO1*/Δ*myo1* strain taken from the heterozygous diploid yeast deletion collection. Deletion alleles of the *YPT11* gene were constructed by amplification of the *HIS3MX6* cassette from plasmid pFA6a-HIS3MX6 [90] using primers 5′ CAC TTG CAT CAG CAT CTC GTA TAT TAG ATA CAT CAT CAA GTC CAT CGG ATC CCC GGG TTA ATT AA and 5′ TGG ACA ATG GCT GCC TGC GAA TCT TGT TGT ATA ATT TGT CGA AGA GAA TTC GAG CTC GTT TAA AC and by amplification of the *hphNT1* cassette from plasmid pFA6a-hphNT1 [91] using primers 5′ GGC ACT TGC ATC AGC ATC TCG TAT ATT AGA TAC ATC ATC AAG TCC ATA TGC GTA CGC TGC AGG TCG AC and 5′ CGT ATT GGA CAA TGG CTG CCT GCG AAT CTT GTT GTA TAA TTT GTC GAA GAC TAA TCG ATG AAT CGA GCT CG.

For genomic insertion of *Su9-DAO*, plasmid pXRnX-Su9-DAO or pXRnX-pGAL-Su9-DAO was linearized with *Asc*I and inserted into the *HO* locus via homologous recombination.

Myo2 was tagged with yEGFP by PCR amplification of the *yEGFP CaURA3* cassette from plasmid pKT209 [83] using primers 5′ AGT TGA CCT TGT TGC CCA ACA AGT CGT TCA AGA CGG CCA CGG TGA CGG TGC TGG TTT A and 5′ TTA GCA TTC ATG TAC AAT TTT GTT TCT CGC GCC ATC AGT TTC GAT GAAT TCG AGC TCG. Genomic tagging of *MMR1* with yEGFP was done by amplification of the *yEGFP CaURA3* cassette from plasmid pKT209 [83] using primers 5′ CCA ACT TAA CCT TCC TGT CCA AGT GGA GAA GAA GGA AAA AGG TGA CGG TGC TGG TTT A and 5′ GTT TGT GTA AAA TAA GTT AAT TTA ATT TGA AGT TGA CGC TTC GAT GAA TTC GAG CTC G.

All yeast strains used in this study are listed in S1 Table.

## Growth and manipulation of yeast

Growth of yeast cultures was always at 30°C. For $H_2O_2$ treatment, logarithmically growing cells in SCD medium (synthetic complete dextrose containing 2% glucose) were treated with 0.3 or 0.5 mM $H_2O_2$ for 15 min and analyzed by fluorescence microscopy. Diamide treatment was done analogously using 1.5 mM diamide. For experiments involving DAO activity, cells were grown in SCD medium without ammonium sulfate containing 0.1% monosodium glutamic acid (MSG). They were cultivated in logarithmic growth phase for at least 24 h. Then, production of matrix $H_2O_2$ was induced by addition of 100 mM D-alanine or 100 mM L-alanine in the control sample. Cells were harvested after 3 h and either analyzed by live cell microscopy or fixed in 3.7% formaldehyde for 30 min at room temperature or analyzed by Western blotting. Counterselection of cells containing *URA3*-based plasmids [92] was by growth on plates containing 1 mg/ml 5-fluoroorotic acid (5-FOA) and 50 μg/ml uracil. Cells containing the p416 (MET25)-GFP-YPT11 plasmid were grown in medium containing 40 mg/l methionine.

To analyze mitochondrial inheritance in zygotes, cells were grown on synthetic complete (SC) medium without ammonium sulfate containing 0.1% MSG with 1% ethanol and 2% glycerol as carbon source for selection of mtDNA maintenance. Cells were then inoculated in SC medium with 2% galactose to allow logarithmic growth overnight. The next morning, cultures were split and incubated with either 100 mM D-alanine or L-alanine. After incubation for 30 min, expression of mitochondrial markers under control of the *GAL* promoter was stopped by addition of 2% of glucose (this step was omitted in assays involving *DNM1 FZO1* wild-type strains). Then, equal amounts of cells from both mating partners were mixed to obtain a culture of 10 ml at an $OD_{600}$ of 0.12. After 3 h of mating, cells were fixed with 3.7% formaldehyde for 30 min. Alternatively, cells were seeded in an ibidi slide (ibidi GmbH, Gräfelfing, Germany) coated with concanavalin A (0.25 mg/ml) for about 10 min, washed 3 times with medium, supplemented with fresh medium containing 100 mM D-alanine or L-alanine, and analyzed by live cell fluorescence microscopy for 2 to 3 h at 30°C.

To analyze mitochondrial inheritance in mitotically dividing diploid cells, haploid cells were grown overnight as described above, equal amounts of cells from both mating partners were mixed to obtain a culture of 10 ml at an $OD_{600}$ of 0.10, and mating was allowed for 3 h at 30°C in the presence of 2% glucose to stop the expression of the mitochondrial markers and the galactose-inducible Su9-DAO construct. Then, either 100 mM D-alanine or L-alanine was added to the culture to induce DAO activity. Cells were incubated for another 3 h to allow proliferation of diploid cells. Cells were either fixed with 3.7% formaldehyde for 30 min or subjected directly to live cell imaging as described above.

## Staining of cellular structures

If not indicated otherwise, staining of cellular structures was always in logarithmically growing cells.

Staining of the actin cytoskeleton was as described [93] with minor modifications. Cells were grown in SCD medium, fixed with 3.7% formaldehyde for 30 min at room temperature, washed in PBS, the actin cytoskeleton was stained for 1 h with rhodamine-phalloidine (Invitrogen) according to the manufacturer's instructions; cells were washed with PBS and analyzed by fluorescence microscopy.

Analysis of mitochondrial membrane potential with TMRM was as described [94] with minor modifications. Cells were grown to logarithmic growth phase in SCD medium, 0.2 OD cells were harvested, washed with washing buffer (10 mM HEPES/KOH (pH 7.2), 5% glucose), stained for 20 min in 1 ml staining buffer (10 mM HEPES/KOH (pH 7.2), 5% glucose, 125 nM TMRM) under agitation, washed twice in PBS, resuspended in washing buffer, and immediately analyzed by fluorescence microscopy.

Vacuoles were stained by addition of 100 μM CellTracker Blue CMAC (7-amino-4-chlormethylcoumarin) to the culture. After incubation for 30 min under agitation, cells were washed two times with PBS, resuspended in medium, and analyzed by fluorescence microscopy.

For PI staining, cells were resuspended in 10 mM HEPES/2% glucose buffer (pH 7.2) and PI was added to a final concentration of 0.2 μg/ml. After incubation for 15 min under agitation, cells were washed with PBS, resuspended in 10 mM HEPES/2% glucose buffer (pH 7.2), and analyzed by fluorescence microscopy.

## Microscopy

Cells were analyzed using a Leica DMi8 fluorescence microscope (Leica Microsystems GmbH, Wetzlar, Germany) equipped with an HC PL APO 100×/1.40 OIL objective, a Lumencor SPECTRA X light source and fluorescence filter sets (FITC ex. 460 to 500 nm, em. 512 to 542 nm; TXR ex. 540 to 580 nm, em. 592 to 668 nm; QUAD-P-T ex. 397 to 413 nm, 484 to 496 nm, 557 to 567 nm, 629 to 645 nm, em. external filter wheel 425 to 475 nm). The microscope was equipped with a sCMOS Leica-DFC9000GT-VSC07400 camera. For microscope settings, image generation, and processing (cropping, maximum intensity projection), the Leica LAS X software (version 3.6.0.20104, Leica Microsystems GmbH, Wetzlar, Germany) was used. Deconvolution of z stacks was carried out by Huygens Deconvolution Software (version 18.10, Scientific Volume Imaging, Hilversum, The Netherlands). For time-resolved 3D microscopy, the incubation chamber on the microscope was preheated to 30˚C, cells were immobilized on ibidi slides coated with concanavalin A, z stacks with a step size of 0.5 μm were taken every 60 s or 3 min over a time frame of up to 3 h and subjected to deconvolution. Differential interference contrast (DIC) images were recorded at the beginning and the end of the experiment. Final image processing, including adjustment of brightness, contrast and background reduction, image rescaling (bicubic interpolation), and the overlay of different channels, was done using Adobe Photoshop CS6 (Adobe Systems) or Image J [95].

Quantification of mitochondrial inheritance in zygotes was as follows: z stacks of ERFP and GFP signals were taken with a step size of 0.213 μm. Regions of interest (ROIs) corresponding to outlines of the zygote and its bud based on a DIC image were defined using Fiji software [96]. For each channel, a mask of the mitochondrial network was generated from maximum intensity projections of z stacks using Li thresholding. This mask and ROIs were used to determine fluorescence signals in sum projections of z stacks. After substraction of background signals, the ratio of the GFP signal in the bud to the GFP signal in the total zygote (including its bud) and the ratio the ERFP signal in the bud to the ERFP signal in the total zygote (including its bud) were calculated. A relative inheritance score was determined by dividing the GFP value by the ERFP value:

Relative inheritance = $(GFP_{bud}: GFP_{zygote}) / (ERFP_{bud}: ERFP_{zygote})$.

### Western blot analysis of Mmr1-GFP levels

WT Su9-DAO and Mmr1-GFP Su9-DAO strains were grown as described above, and cell extracts were prepared by alkali treatment and boiling in electrophoresis sample buffer as described [97] with minor modifications (i.e., alkaline treatment was for 10 min on ice). SDS PAGE and blotting to nitrocellulose membranes was performed according to standard procedures. Mmr1-yEGFP was detected using mouse monoclonal antibodies recognizing GFP (Roche Life Sciences, catalog number 11814460001) and secondary antibodies coupled to horseradish peroxidase (Sigma-Aldrich, catalog number A4416). Hexokinase, Hxk1, served as a loading control and was detected using polyclonal rabbit antibodies (Biotrend Biochemikalien, catalog number 100–4159) and secondary antibodies coupled to horseradish peroxidase (Promega, catalog number W4011).

### Statistical analysis

For statistics, SigmaPlot V14 (Systat Software, Erkrath, Germany) was applied. For comparing two groups, unpaired two-tailed Student's $t$ tests with ad hoc normality (Shapiro–Wilk) and equal variance (Browne–Forsythe) tests were applied. If ad hoc equal variance tests failed, unpaired two-tailed Welch's $t$ tests were performed. For all other comparisons, one-way analysis of variance (one-way ANOVA) with ad hoc normality (Shapiro–Wilk) and equal variance (Browne–Forsythe) tests and Holm–Sidak as post hoc test were applied. Differences were considered to be significant with $p$ values <0.05 (*), <0.01 (**), and <0.001 (***). Error bars indicate standard deviation (SD) of mean values of biological replicates collected in independent experiments. Data pooling and statistics for each experiment is detailed in S2 Table.

### Supporting information

**S1 Fig. $H_2O_2$-treated cells remain viable and diamide treatment induces reversible fragmentation of mitochondria; extended data related to Fig 1.** **(A)** Yeast cells were treated with 0.5 mM $H_2O_2$ for 15 min, stained with PI, and analyzed by DIC and fluorescence microscopy. Cells that were killed at 100˚C served as a positive control. Images were taken with identical camera settings. Bar, 10 μm. A quantification is shown in Fig 1C. **(B and C)** Yeast cells expressing mtGFP were treated with exogenously added diamide for 30 min and analyzed by DIC and fluorescence microscopy. For recovery cells were incubated for 90 min in fresh medium. Fluorescence images are z stacks subjected to deconvolution. Mitochondrial morphology was quantified in 150 cells per sample (triplicate experiments ± SD). Bar, 5 μm. The data underlying this figure can be found in S1 Datasheet.
(TIF)

**S2 Fig. Mitochondrially produced $H_2O_2$ induces loss of membrane potential, mitochondrial fragmentation, and block of fusion; extended data related to Fig 2.** **(A)** Wild-type cells, cells lacking mtDNA ($rho^0$), and wild-type cells expressing Su9-DAO were incubated without or with L-alanine or D-alanine for 3 h and then stained with TMRM. Fluorescence microscopy images are maximum intensity projections of z stacks using identical camera settings. Bar, 5 μm. **(B)** Wild-type cells expressing Su9-DAO were incubated in the presence of L-alanine or D-alanine for 3 h and stained with PI. Heat-killed cells were incubated for 5 min at 100˚C. Fluorescence images were taken with identical camera settings. Bar, 10 μm. **(C and D)** Wild-type cells expressing mtGFP and lacking (left) or expressing (right) Su9-DAO were incubated for 3 h in the presence of L-alanine or D-alanine. Fluorescence images are z stacks subjected to deconvolution. Mitochondrial morphology was quantified in 100 cells per sample (3 biological replicates ± SD). Bar, 5 μm. **(E)** Wild-type cells expressing either Su9-DAO together

with mtGFP (left) or only mtGFP (right) and wild-type cells expressing Su9-DAO together with mtERFP were incubated for 30 min with L-alanine or D-alanine, mixed, and incubated for another 3 h to allow mating and zygote formation. Fluorescence images are z stacks subjected to deconvolution. Single channel images shown here were used to generate merged images shown in Fig 2H. Bar, 5 μm. The data underlying this figure can be found in S1 Datasheet.
(TIF)

**S3 Fig. Δ*myo1* cells are misshapen and show an abnormal budding pattern.** Logarithmically growing cells in minimal complete medium containing glucose as carbon source were analyzed by DIC microscopy. Bar, 5 μm.
(TIF)

**S4 Fig. The *MMR1-yEGFP* allele is functional; extended data related to Fig 6.** Strain Δ*ypt11* (strain no. 19 in S1 Table) and a wild-type strain carrying a genomic insertion of the *MMR1-yEGFP* allele at the *MMR1* locus (strain no. 15 in S1 Table) were mated, and the resulting diploid strain was subjected to sporulation and tetrad dissection. Ascospores were allowed to grow to colonies on YPD plates. Four colonies in each column correspond to 1 tetrad. Note that all 7 clones carrying both the Δ*ypt11* and *MMR1-yEGFP* alleles grow like wild type. Compare Fig 7D for the lethal phenotype of Δ*mmr1* Δ*ypt11* in the genetic background used in this study.
(TIF)

**S5 Fig. Healthy mitochondria are selectively inherited in buds formed at zygotes; extended data related to Fig 8.** (A) The experiment was performed as in Fig 8, but with switched mating types. (B) Fluorescence images are z stacks subjected to deconvolution. Asterisks indicate medial buds of zygotes. Bar, 5 μm. (C) Relative mitochondrial inheritance was quantified in 50 D-alanine-treated zygotes by calculating the GFP and ERFP intensity ratios as shown in the equation. The arrow points to the zygote that is closest to the value of 1.
(TIF)

**S1 Video. Mitochondrial dynamics in wild-type yeast cells expressing mtGFP (control conditions).** Z stacks with a step size of 0.5 μm were taken every 60 s, total time 30 min, time lapse shows 2 frames per second; z stacks were subjected to deconvolution, DIC images were taken at the beginning and the end of the experiment. The video is related to Fig 1A and 1B.
(MP4)

**S2 Video. Mitochondrial dynamics in wild-type yeast cells expressing mtGFP treated with 0.5 mM H₂O₂.** Z stacks with a step size of 0.5 μm were taken every 60 s, total time 30 min, time lapse shows 2 frames per second; z stacks were subjected to deconvolution, DIC images were taken at the beginning and the end of the experiment. The video is related to Fig 1A and 1B.
(MP4)

**S3 Video. Mitochondrial dynamics in wild-type yeast cells expressing mtGFP recovering from H₂O₂ treatment.** Cells were incubated for 15 min with 0.3 mM H₂O₂, washed 3 times with medium, and then processed for imaging. Z stacks with a step size of 0.5 μm were taken every 3 min, total time 90 min, time lapse shows 2 frames per second; z stacks were subjected to deconvolution, DIC images were taken at the beginning and the end of the experiment. The video is related to Fig 1A and 1B.
(MP4)

**S4 Video. Mitochondrial dynamics in wild-type yeast cells expressing mtGFP and Su9-DAO in the presence of 100 mM L-alanine (control conditions).** Z stacks with a step size of 0.5 μm were taken every 3 min, total time 180 min, time lapse shows 2 frames per second; z stacks were subjected to deconvolution, DIC images were taken at the beginning and the end of the experiment. The video is related to Figs 2C, 2D, 3A, and 3B.
(MP4)

**S5 Video. Mitochondrial dynamics in wild-type yeast cells expressing mtGFP and Su9-DAO in the presence of 100 mM D-alanine.** Z stacks with a step size of 0.5 μm were taken every 3 min, total time 180 min, time lapse shows 2 frames per second; z stacks were subjected to deconvolution, DIC images were taken at the beginning and the end of the experiment. The video is related to Figs 2C, 2D, 3A, and 3B.
(MP4)

**S6 Video. Mitochondrial inheritance in Δdnm1 Δfzo1 x Δdnm1 Δfzo1 zygotes that contain mitochondria preloaded with either mtERFP only (magenta) or Su9-DAO together with mtGFP (green) in the presence of 100 mM L-alanine (control conditions).** Z stacks with a step size of 0.5 μm were taken every 3 min, total time 120 min, time lapse shows 2 frames per second; z stacks were subjected to deconvolution, DIC images were taken at the beginning and the end of the experiment. The video is related to Fig 8.
(MP4)

**S7 Video. Mitochondrial inheritance in Δdnm1 Δfzo1 x Δdnm1 Δfzo1 zygotes that contain mitochondria preloaded with either mtERFP only (magenta) or Su9-DAO together with mtGFP (green) in the presence of 100 mM D-alanine.** Z stacks with a step size of 0.5 μm were taken every 3 min, total time 120 min, time lapse shows 2 frames per second; z stacks were subjected to deconvolution, DIC images were taken at the beginning and the end of the experiment. The video is related to Fig 8.
(MP4)

**S8 Video. Mitochondrial inheritance in homozygous diploid Δdnm1 Δfzo1 cells that contain mitochondria preloaded with either mtERFP only (magenta) or Su9-DAO together with mtGFP (green) in the presence of 100 mM L-alanine (control conditions).** Z stacks with a step size of 0.5 μm were taken every 3 min, total time 120 min, time lapse shows 2 frames per second; z stacks were subjected to deconvolution, DIC images were taken at the beginning and the end of the experiment. The video is related to Fig 9.
(MP4)

**S9 Video. Mitochondrial inheritance in homozygous diploid Δdnm1 Δfzo1 cells that contain mitochondria preloaded with either mtERFP only (magenta) or Su9-DAO together with mtGFP (green) in the presence of 100 mM D-alanine.** Z stacks with a step size of 0.5 μm were taken every 3 min, total time 120 min, time lapse shows 2 frames per second; z stacks were subjected to deconvolution, DIC images were taken at the beginning and the end of the experiment. The video is related to Fig 9.
(MP4)

**S1 Table. Yeast strains used in this study.**
(PDF)

**S2 Table. Data pooling and statistics.**
(PDF)

**S1 Datasheet. Numerical values underlying summary data in Figs 1B, 1C, 1E, 1G, 2D, 2F, 2I, 3B, 3D, 3F, 3H, 4B, 4E, 5B, 5D, 5F, 6B, 6C, 6E, 7B, 7F, 8C, 9C, S1C, S2D, and S5C.**
(XLSX)

**S1 Raw images. Uncropped blot images in Fig 6F.**
(PDF)

## Acknowledgments

We are grateful to the Westermann and Klecker laboratory for helpful discussions and Stefan Böckler and the students Leonie Scheder, Youli Stepanov, Markus Wehrmann, and Verena Geiger for their contributions to the project. We thank Jan Riemer (Universität zu Köln, Germany), Agnieszka Lewandowska, and Janet Shaw (University of Utah, USA) for making plasmids available to us.

## Author Contributions

**Conceptualization:** Xenia Chelius, Till Klecker, Benedikt Westermann.

**Formal analysis:** Ralf J. Braun.

**Funding acquisition:** Benedikt Westermann.

**Investigation:** Xenia Chelius, Veronika Bartosch, Nathalie Rausch, Magdalena Haubner.

**Supervision:** Till Klecker, Benedikt Westermann.

**Visualization:** Xenia Chelius, Veronika Bartosch, Nathalie Rausch, Jana Schramm, Till Klecker.

**Writing – original draft:** Benedikt Westermann.

**Writing – review & editing:** Xenia Chelius, Veronika Bartosch, Nathalie Rausch, Magdalena Haubner, Jana Schramm, Ralf J. Braun, Till Klecker, Benedikt Westermann.

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
