## [Editor Report · Decision Letter 0]

5 Apr 2023

Dear Dr Westermann, 

Thank you for submitting your manuscript entitled "Selective retention of dysfunctional mitochondria during asymmetric cell division in yeast" for consideration as a Research Article by PLOS Biology. Please accept my apologies for the delay in getting back to you as we consulted with an academic editor about your submission. 

Your manuscript has now been evaluated by the PLOS Biology editorial staff, as well as by an academic editor with relevant expertise, and I am writing to let you know that we would like to send your submission out for external peer review.

Once your full submission is complete, your paper will undergo a series of checks in preparation for peer review. After your manuscript has passed the checks it will be sent out for review. To provide the metadata for your submission, please Login to Editorial Manager (https://www.editorialmanager.com/pbiology) within two working days, i.e. by Apr 07 2023 11:59PM.

Kind regards,

Richard

Richard Hodge, PhD

Associate Editor, PLOS Biology

rhodge@plos.org

PLOS

---

## [Decision Letter · Decision Letter 1]

10 May 2023

Dear Dr Westermann,

Thank you for your patience while your manuscript "Selective retention of dysfunctional mitochondria during asymmetric cell division in yeast" was peer-reviewed at PLOS Biology. Please accept my apologies for the delays that you have experienced during the peer review process. Your manuscript has now been evaluated by the PLOS Biology editors, an Academic Editor with relevant expertise, and by three independent reviewers. 

In light of the reviews, which you will find at the end of this email, we would like to invite you to revise the work to thoroughly address the reviewers' reports.

As you will see, the reviewers find your manuscript interesting and well done. Reviewer’s #2 (point 1) and #3 (point 5) suggest some additional experiments to enhance the overall message of the paper and to rule out alternative interpretations. After discussions with the Academic Editor, we will not make the inclusion of this additional data essential to consider a revised version, but we do encourage you to perform these experiments to confirm and strengthen the significance of the findings. 

Given the extent of revision needed, we cannot make a decision about publication until we have seen the revised manuscript and your response to the reviewers' comments. Your revised manuscript is likely to be sent for further evaluation by all or a subset of the reviewers.

**IMPORTANT - SUBMITTING YOUR REVISION**

*Re-submission Checklist*

*Published Peer Review*

*PLOS Data Policy*

*Blot and Gel Data Policy*

Sincerely,

Richard

Richard Hodge, PhD

Associate Editor, PLOS Biology

rhodge@plos.org

REVIEWS:

Reviewer #1: In the manuscript entitled "Selective retention of dysfunctional mitochondria...", Westermann and co-workers report the establishment of a method to selectively induce ROS formation in yeast mitochondria and use this tool to study the consequences that the resulting damage has on mitochondrial inheritance. They show that damage causes the fragmentation and retention of the affected mitochondria in the yeast mother cell, but not of unaffected mitochondria in the same cell, and dissect the mechanisms involved in decoupling the damaged mitochondria from the inheritance machinery. They show that especially the type V myosin/Myo2 receptor Mmr1 dissociates from damaged mitochondria, probably preventing the recruitment of Myo2, and hence stopping the motility of these organelles and their movement towards the bud.

Overall, this is a beautiful study for several reasons. First, the notion that "unfit" mitochondria are prevented from being inherited by the bud has been around for a while but the lack of appropriate assays and tools to study this idea rigorously had prevented from substantiating and confirming the idea in a definitive manner. Second, given these experimental limitations it had remained extremely difficult to acquire any mechanistic insights into how this may function. The manuscript of Westermann and colleagues addresses these two points in a very rigorous and definitive manner, establishing a very robust assay and using it to identify the key players involved. The data is very solid, convincingly conclusive and the experiments are extremely well controlled. The text is clear and reads well. The insights gained are very interesting and important. Therefore, I enthusiastically recommend publishing this very elegant study in your columns.

I see only one element that the authors might want to address before publication: The authors report that mitochondrial defects affect also vacuole inheritance (Figure 3G). However, the picture shown in that panel do not convey that message. Out of four cells shown in D-Ala, three show proper vacuolar inheritance. Interestingly, the affected cell does not show mitochondrial retention in the mother cell (mitochondrial signal is clearly seen in the bud). Perhaps, the authors could comment or show more representative pictures?

Reviewer #2: In the manuscript by Chelius et al, the authors describe an investigation into the mechanism of selective mitochondrial retention during budding and mitosis in S. cerevisiae. They first demonstrate that yeast mitochondria respond to H2O2 by fragmentation, as previously reported by others, and by loss of motility due to loss of actin cable structures. To separate mitochondria-specific responses from indirect effects on other cellular structures the authors express mitochondrially targeted D-amino acid oxidase. They demonstrate that mitochondrially targeted DAO causes mitochondrial dysfunction and morphological fragmentation in response to D-amino acid supplementation. They demonstrate that this is independent of oxphos activity and show that the fragmentation response is a result of inhibition of mitochondrial fusion. In addition, this also inhibited mitochondrial transmission to the bud during mitosis. These appear to be mitochondria-specific as actin cables remained intact and myo2 localization was not affected. The authors then ask whether the fragmentation phenotype is linked to the bud transport phenotype, by testing whether it is mitochondrial size that affects bud transport, or dynamics. They find that they can uncouple the inheritance phenotype from the fragmentation phenotype, by using double dnm1D fzo1D cells. Importantly they demonstrate that increasing Myo2 attachment to mitochondria by over expression of Ypt11 or Mmr1 can suppress the mitochondrial inheritance defect of mitochondrial DAO activity. Somewhat paradoxically, deletion of MMR1, but nor YPT11, also rendered mitochondrial inheritance independent of DAO expression. In agreement with a role for these proteins in regulating Myo2 association with mitochondria, an artificially mitochondrial-tethered Myo2 construct also rendered cells DAO- insensitive. 

The authors then carry out a series of experiments in artificially heteroplasmic diploids containing separate noninteracting pools of DAO positive and DAO negative mitochondria showing that cells can select the DAO negative mitochondria for inheritance, and that Mmr1 is required for this selection. 

Overall, this is an elegant paper that contains well thought out experiments and sheds new light on the enigmatic process of selective mitochondrial inheritance. Nonetheless, I find a number of weaknesses that I suggest the authors address before considering publication. 

First, the authors are using the mt-DAO construct to selectively generate H2O2 in mitochondria. This is an elegant approach, which seems to indicate that the target for H2O2 action is mitochondrial, and suggests some kind of signaling mechanism that would then regulate the cytoplasmic factors such as Ypt11/Mmr1. However the authors do not rule out the possibility that mitochondrial H2O2 is simply leaking out and directly activating a cytoplasmic response. The authors could address this by asking whether overexpression of cytosolic versus mitochondrial catalase can suppress the effects of DAO. 

A second weakness is that I see, is that overexpression of Mmr1 suppresses the effects of mtDAO, but so does the deletion of Mmr1! I think the authors must address this 'paradox', whether experimentally or in the discussion. 

Reviewer #3: In this manuscript the authors have studied the impact of oxidative damage originating in mitochondria on mitochondrial inheritance during cell division in yeast. This was performed by expressing D-amino oxidase in mitochondrial matrix, and incubation of cells with the substrate D-alanine or L-alanine as a control. The study is interesting and the most relevant observations are: a) mitochondrial oxidative stress causes specific alterations of the affected mitochondria, and not of the normal mitochondrial present in the same cell, and b) oxidative stress causes the dislocalization of Mmr1 in ROS-stressed mitochondria. The manuscript shows a number of weak aspects that require some attention.

1. The 10-fold serial dilution experiments shown in the manuscript should be quantified (such as in figures 1A and 1G). Why L-alanine impacts on cells grown in glycerol plus ethanol? Please, provide an explanation. 

2. During the manuscript the authors indicate that mitochondrially produced ROS enhances the percentage of cells that do not inherit any mitochondria during division. This is mentioned in figure 1B but in others. The authors should provide an explanation for that and specifically refer to this in the rest of the figures.

3. In Figure 4, the authors should document the impact of just depletion of Fzo1 on mitochondrial inheritance. Otherwise they cannot conclude that the effects of ROS production are independent of mitochondrial fusion.

4. Data from Figure 6 suggest that Mmr1 disappear from the outer mitochondrial membrane of those mitochondria producing ROS. Is that due to degradation of the protein or is a consequence of dissociation of Mmr1 from mitochondria? Do you see alterations in the total content of Mmr1 in cells upon ROS production? This is relevant in order to understand the process that takes place under ROS conditions, and also based on data by Obara et al indicating an active proteolysis of Mmr1 protein (PMID: 35422486). 

5. A basic question not addressed in the manuscript is whether there are differences in mitochondrial respiration between daughter and mother cells under conditions of high mitochondrial ROS production. This missing piece of evidence would validate the concept of a retention of damaged mitochondria in mother cells, and it would enhance the message of the manuscript.

---

## [Editor Report · Decision Letter 2]

23 Aug 2023

Dear Dr Westermann,

Thank you for your patience while we considered your revised manuscript "Selective retention of dysfunctional mitochondria during asymmetric cell division in yeast" for publication as a Research Article at PLOS Biology. This revised version of your manuscript has been evaluated by the PLOS Biology editors and the Academic Editor. Thank you for already providing the underlying data for the main and supplementary figures and the original western blot images.

Based on our Academic Editor's (Jonathon Pines) assessment of your revision, I am pleased to say that we can accept your manuscript for publication, provided you address any remaining formatting and reporting issues. These will be detailed in an email you should receive within 2-3 business days from our colleagues in the journal operations team; no action is required from you until then. Please note that we will not be able to formally accept your manuscript and schedule it for publication until you have completed any requested changes.

During the production process, please take this last chance to review your reference list to ensure that it is complete and correct. If you have cited papers that have been retracted, please include the rationale for doing so in the manuscript text, or remove these references and replace them with relevant current references. Any changes to the reference list should be mentioned in the cover letter that accompanies your revised manuscript.

PRESS

Best wishes, 

Richard

Richard Hodge, PhD

rhodge@plos.org

PLOS
